# AsFT: Anchoring Safety During LLM Fine-Tuning Within Narrow Safety Basin

## Abstract

Large language models (LLMs) are vulnerable to safety risks during fine-tuning, where even small amounts of malicious or benign data can compromise safeguards. In this paper, building on the concept of the *alignment direction*—defined by the weight difference between aligned and unaligned models—we observe that perturbations along this direction preserve model safety. In contrast, perturbations orthogonal to this alignment are strongly correlated with harmful updates, rapidly degrading safety and framing the parameter space as a "narrow safety basin." Based on this insight, we propose **AsFT** (Anchoring Safety in Fine-Tuning), a data-free method that formulates safety-preserving fine-tuning as a constrained optimization problem. AsFT uses the alignment direction as an anchor and restricts parameter updates within the "narrow safety basin" through a tractable Lagrangian relaxation, thereby suppressing harmful updates while preserving task-relevant adaptation. Extensive experiments across multiple datasets and models demonstrate that AsFT reduces harmful behaviors by up to 7.60%, improves task performance by 3.44%, and consistently outperforms existing methods across diverse fine-tuning scenarios.

## 1 Introduction

The rapid advancement of large language models (LLMs) has led to their widespread adoption across various industries, where fine-tuning is essential to adapt these models to specific tasks and scenarios. However, fine-tuning exposes critical safety vulnerabilities. Even small amounts of malicious or harmless data during fine-tuning can compromise the model's safeguards, causing the models to generate harmful outputs post-fine-tuning [25, 5, 44]. This raises the urgent need for methods that balance task-specific utility with robust safety defenses [26].

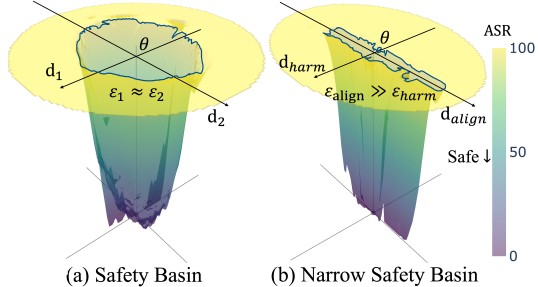

(a) Safety Basin  (b) Narrow Safety Basin

Figure 1: (a) The Safety Basin [42] shows a region where perturbations along $d_{\mathrm{random}}$ preserve model safety, while safety sharply declines outside this area. (b) The Narrow Safety Basin demonstrates the asymmetry between $d_{\mathrm{aligned}}$ and $d_{\mathrm{harm}}$, where $d_{\mathrm{aligned}}$ allows larger perturbations, while $d_{\mathrm{harm}}$ causes sharp safety declines. In both subfigures, **lower values** indicate **higher safety**.

Currently, there are various strategies for enhancing safety during LLM fine-tuning. While these strategies primarily rely on data-driven methods, they face a significant challenge: reliance on high-quality datasets, which are both costly and susceptible to bias [26]. Post-tuning methods like Safe LoRA [19] mitigate fine-tuning's negative impact on model safety by discretizing and projecting LoRA weights into a safety-aligned subspace. However, they overlook layer continuity, as discrete projections can disrupt the consistency of learned features across layers. By focusing primarily on safety-related features, they neglect the performance-related characteristics brought by training data, degrading models' performance.

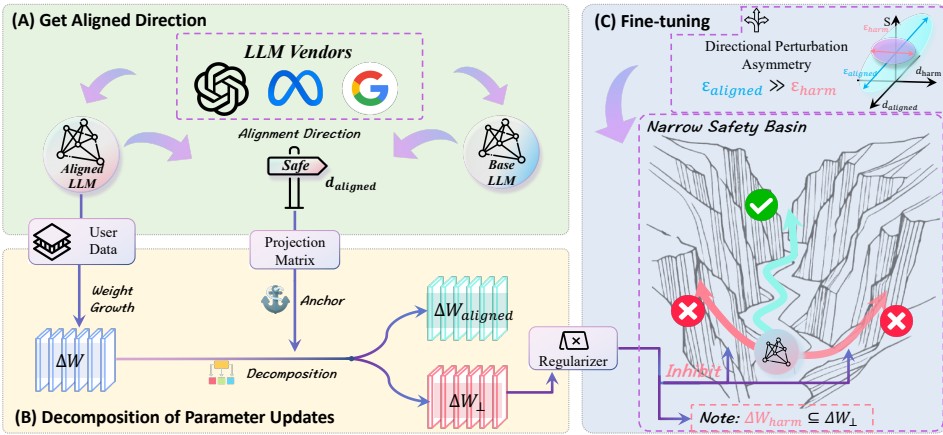

Figure 2: The proposed AsFT decomposes parameter updates into $d_{\text{aligned}}$ and $d_{\perp}$, suppresses harmful updates along $d_{\perp}$ by regularization and constraining updates within the narrow safety basin.

To address the limitations mentioned above, we aim to develop a data-free approach that leverages continuous optimization to enhance safety during fine-tuning. We observe that aligned models (e.g., Llama-Chat), developed under rigorous protocols, exhibit robust defenses against harmful inputs [44, 19], whereas their unaligned counterparts (i.e., base models) lack such safeguards. This contrast inspires us to explore the latent information within the model parameter space. The weight difference between these two models encapsulates the alignment efforts undertaken by LLM vendors to enhance model safety. It not only reflects the core alignment process but also provides a critical direction for safety optimization [19, 8, 68].

Given these observations, this paper hypothesizes that the alignment direction can guide safety-preserving updates during fine-tuning and thus addresses the following question:

*Can this weight difference serve as an anchor to guide safety-preserving updates?*

Following prior work on safety landscape [42] (Figure 1(a)), we define the alignment direction ($d_{\text{aligned}}$) based on this weight difference $\Delta \mathbf{W}$ and observe that perturbations along $d_{\text{aligned}}$ effectively preserve model's safety. Conversely, directions orthogonal to $d_{\text{aligned}}$ (denoted as $d_{\perp}$) are strongly correlated with harmful directions, where even small perturbations along $d_{\perp}$ can rapidly and significantly compromise the model's safety. This conceptualization frames the LLM parameter space as a "narrow safety basin" (as shown in Figure 1(b)), within which model's safety can be preserved by guiding updates along the constrained region defined by $d_{\text{aligned}}$.

Leveraging this insight, we propose AsFT (as shown in Figure 2), a novel method that anchors safety during fine-tuning by explicitly guiding parameter updates within the confines of a "narrow safety basin". While the exact harmful direction ($d_{\text{harm}}$) is generally inaccessible, we use $d_{\perp}$, derived from $d_{\text{aligned}}$, as a proxy to approximate and suppress harmful parameter updates. AsFT frames fine-tuning as a constrained optimization problem, where updates are restricted to remain close to the alignment subspace. We implement this constraint through a tractable relaxation that continuously suppresses deviations along $d_{\perp}$, effectively preserving the safety of the fine-tuned model while maintaining strong task-specific performance. Experimental results demonstrate that AsFT reduces harmful scores by up to $7.60\%$ compared to Safe LoRA, while delivering superior performance on a variety of downstream tasks. In summary, our contributions are as follows:

- We observe that the alignment direction $d_{\text{aligned}}$ can serve as a safety anchor and that its orthogonal counterpart $d_{\perp}$ closely aligns with the harmful direction $d_{\text{harm}}$, framing the LLM safety landscape as a "narrow safety basin".
- We propose AsFT (Anchoring Safety in Fine-Tuning), which suppresses parameter updates along $d_{\perp}$, enabling fine-tuning within the "narrow safety basin" to preserve alignment safety.
- We validate AsFT through extensive experiments across multiple models, tasks, and fine-tuning attacks, achieving notable improvements in both safety and downstream task performance.

## 2  Related Work

Safety alignment ensures that large language models (LLMs) generate outputs aligned with human values and ethics [53, 4, 69, 16, 17, 62, 12]. Key techniques include instruction fine-tuning [56], RLHF[41], DPO [46], and others. However, these methods are vulnerable to small-scale fine-tuning attacks, where minimal harmful or neutral data can compromise model safety [44, 58]. To address this, defenses have been developed across three stages: alignment, fine-tuning, and post-tuning [22]. **Full related work** and **key differences** (e.g., Task [27] and Refusal [3] Vector and Safe LoRA [19]) are provided in Appendix E.2 and E.3.

**Alignment Phase Defenses** aim to fortify models against harmful fine-tuning attacks by enhancing robustness during the alignment phase [43, 65, 37]. Methods like Vaccine [23] introduce latent perturbations in the parameter space to ensure aligned outputs under adversarial conditions, while RepNoise [47] eliminates harmful representations to effectively prevent their reconstruction. TAR [51] optimizes parameters to sustain high harmful loss even after adversarial fine-tuning, and Booster [25] minimizes the drop in harmful loss under simulated attacks. T-Vaccine [35] further strengthens these defenses by selectively perturbing safety-critical model layers.

**Fine-tuning Phase Defenses** enhance safety during training to counter harmful fine-tuning [40, 55, 2, 32]. MLLR [13] identifies safety-critical modules via modular robustness analysis and applies differential learning rates. SafeInstr [5] incorporates safety-focused examples. Lisa [21] limits optimization drift using dual-state optimization with alignment data and proximity constraints. BEA [54] embeds hidden triggers to suppress harmful content. Seal [48] excludes harmful samples via two-stage optimization. SAFT [9] filters harmful data by subspace decomposition-based scoring.

**Post-tuning Phase Defenses** aim to restore model safety after harmful fine-tuning attacks [7]. Safe LoRA [19] discretely projects parameters onto the safe direction after fine-tuning. SOMF [61] integrates additional benign task knowledge and reuses essential safety parameters. Antidote [24] effectively prunes harmful parameters during the post-processing stage, and SafetyLock [68] leverages extracted safety directions to actively intervene in attention head activations during inference.

## 3  Methodology

### 3.1  Preliminaries: Safety Landscape and Basin

The Safety Landscape, introduced by Peng et al. [42], characterizes how LLMs' safety varies across their parameter space, evaluated using a monotonic function $S(\cdot)$, where lower values indicate higher safety, typically measured as the Attack Success Rate (ASR). Let $\theta$ denote model weights, $d$ the perturbation direction, and $\alpha$ the perturbation magnitude, with $\hat{d} = d/|d|$ as a normalized direction. For two orthogonal directions, the safety landscape is defined as:

$$f(\alpha, \beta) = S(\theta + \alpha\hat{d}_1 + \beta\hat{d}_2). \tag{1}$$

In this context, Peng et al. [42] identified the concept of a Safety Basin (Fig. 1(a), with drawing details provided in Appendix D.2). Therefore, we formalize this concept as $\mathcal{B}(\theta; \epsilon_1, \epsilon_2)$, which refers to a localized region in the parameter space where the model's safety remains robust against bounded perturbations, within the limits defined by the maximum allowable perturbations $\epsilon_1$ and $\epsilon_2$:

**Definition 1 (Safety Basin)** *The Safety Basin, denoted as $\mathcal{B}(\theta; \epsilon_1, \epsilon_2)$, is formally defined as*

$$\mathcal{B}(\theta; \epsilon_1, \epsilon_2) = \Big\{ (\alpha, \beta) \in \mathbb{R}^2 \,\Big|\, S(\theta + \alpha\hat{d}_1 + \beta\hat{d}_2) \leq S_{threshold},$$
$$|\alpha| \leq \epsilon_1, \, |\beta| \leq \epsilon_2 \Big\}. \tag{2}$$

### 3.2  Rethinking the Safety Basin

The original Safety Basin concept [42] implicitly assumes isotropy in the parameter space, i.e., that perturbations along random directions affect safety uniformly. However, this assumption overlooks

Table 1: Cosine Similarity between harmful direction ($d_{\text{harm}}$) and alignment direction ($d_{\text{aligned}}$), along with the effective rank of $d_{\text{harm}}$ evaluated across multiple harmful datasets [49, 70, 28, 39]. For reference, the cosine similarity between $d_{\text{random}}$ and $d_{\text{aligned}}$ is $8.488 \times 10^{-3}$, which is substantially higher than the average $d_{\text{harm}}$ of $6.30 \times 10^{-4}$. Detailed results are provided in Appendix Table 27.

| Number of Samples | Harmful [49] | | BeaverTails [70] | | AdvBench [28] | | HarmBench [39] | | Average | |
|---|---|---|---|---|---|---|---|---|---|---|
| | Cos. Sim. | Eff.Rank | Cos. Sim. | Eff.Rank | Cos. Sim. | Eff.Rank | Cos. Sim. | Eff.Rank | Cos. Sim. | Eff.Rank |
| 10 | $7.12 \times 10^{-4}$ | 156.64 | $9.12 \times 10^{-5}$ | 215.92 | $7.68 \times 10^{-4}$ | 130.86 | $8.09 \times 10^{-4}$ | 153.15 | $5.95 \times 10^{-4}$ | 164.14 |
| 20 | $7.40 \times 10^{-4}$ | 146.13 | $1.10 \times 10^{-4}$ | 234.66 | $7.47 \times 10^{-4}$ | 126.40 | $6.71 \times 10^{-4}$ | 156.67 | $5.67 \times 10^{-4}$ | 165.96 |
| 50 | $6.46 \times 10^{-4}$ | 197.89 | $9.00 \times 10^{-5}$ | 265.14 | $8.61 \times 10^{-4}$ | 123.26 | $7.87 \times 10^{-4}$ | 184.12 | $5.96 \times 10^{-4}$ | 192.60 |
| 100 | $1.18 \times 10^{-3}$ | 212.51 | $1.46 \times 10^{-4}$ | 291.02 | $8.48 \times 10^{-4}$ | 132.26 | $7.39 \times 10^{-4}$ | 145.85 | $7.28 \times 10^{-4}$ | 195.41 |
| 200 | $9.92 \times 10^{-4}$ | 177.56 | $1.26 \times 10^{-4}$ | 226.08 | $9.14 \times 10^{-4}$ | 132.61 | $7.17 \times 10^{-4}$ | 149.03 | $6.87 \times 10^{-4}$ | 171.32 |
| 500 | $8.56 \times 10^{-4}$ | 220.84 | $8.83 \times 10^{-5}$ | 222.58 | $7.43 \times 10^{-4}$ | 132.98 | $7.33 \times 10^{-4}$ | 171.30 | $6.05 \times 10^{-4}$ | 186.93 |
| **Average** | $\mathbf{8.54 \times 10^{-4}}$ | **185.26** | $\mathbf{1.09 \times 10^{-4}}$ | **242.57** | $\mathbf{8.14 \times 10^{-4}}$ | **129.73** | $\mathbf{7.43 \times 10^{-4}}$ | **160.02** | $\mathbf{6.30 \times 10^{-4}}$ | **179.39** |

the fact that modern alignment processes are highly structured interventions: alignment is not random noise added to model parameters, but rather a targeted transformation guided by reinforcement learning from human feedback (RLHF) or instruction tuning. This raises a critical question: does the parameter space truly exhibit uniform safety properties, or does the alignment process itself induce an anisotropic geometry where certain directions are safety-preserving while others are safety-vulnerable?

To probe this question, we focus on the alignment direction, defined as

$$d_{\text{aligned}} = \theta_{\text{aligned}} - \theta_{\text{unaligned}}, \tag{3}$$

which captures the essential weight transformations introduced by alignment. If isotropy held, perturbations orthogonal to $d_{\text{aligned}}$ should not systematically correlate with safety degradation.

**Empirical Evidence of Anisotropy.** To test this hypothesis, we compare the cosine similarity between $d_{\text{aligned}}$ and update directions induced by harmful ($d_{\text{harm}}$) and random ($d_{\text{random}}$) fine-tuning. We fine-tuned Llama-2-7B with varying amounts of harmful data (10–500 samples across multiple datasets [49, 70, 28, 39]), deriving $d_{\text{harm}}$ as the difference between the harmful fine-tuned model and the original aligned model. For comparison, we also constructed random directions $d_{\text{random}}$ sampled uniformly in parameter space.

As shown in Table 1, $d_{\text{harm}}$ is nearly orthogonal to $d_{\text{aligned}}$, with cosine similarity consistently at $10^{-4}$. In contrast, random directions exhibit higher similarity at the $10^{-3}$ level. This order-of-magnitude difference demonstrates that $d_{\text{harm}}$ is not merely random variation but occupies a distinct subspace strongly separated from $d_{\text{aligned}}$. Further, analyzing the effective rank [11] of harmful update directions reveals a low-dimensional structure: while the full parameter rank of Llama-2-7B is $\approx 4000$, the average effective rank of $d_{\text{harm}}$ is only 179.39. This indicates that harmful updates are confined to a narrow, low-rank subspace, further supporting anisotropy in the safety landscape.

**Anisotropy of the Safety Landscape.** Figure 1(b) visualizes safety sensitivity along $d_{\text{aligned}}$ versus $d_{\text{harm}}$. Perturbations along $d_{\text{aligned}}$ preserve safety, while even small perturbations along $d_{\text{harm}}$ sharply degrade it. The asymmetry in tolerable perturbation ranges ($\epsilon_{\text{aligned}} \gg \epsilon_{\text{harm}}$) confirms that safety is highly directional: the model is robust to movement along the alignment direction but acutely vulnerable along the harmful direction.

**Narrow Safety Basin.** We thus refine the original Safety Basin definition and introduce the *Narrow Safety Basin*, where safety robustness is dominated by anisotropy between $d_{\text{aligned}}$ and $d_{\text{harm}}$:

**Definition 2 (Narrow Safety Basin)** *The Narrow Safety Basin, $\mathcal{B}_{narrow}(\theta; \epsilon_1, \epsilon_2)$, satisfies:*

$$\mathcal{B}_{narrow}(\theta; \epsilon_1, \epsilon_2) = \Big\{ (\alpha, \beta) \in \mathbb{R}^2 \,\Big|\, S(\theta + \alpha \hat{d}_{aligned} + \beta \hat{d}_{harm}) \leq S_{threshold},$$
$$|\alpha| \leq \epsilon_1, \, |\beta| \leq \epsilon_2, \, \epsilon_1 \gg \epsilon_2 \Big\}. \tag{4}$$

where, $\epsilon_1 \gg \epsilon_2$ indicates that the allowable perturbation range along $d_{\text{aligned}}$ is much larger than $d_{\text{harm}}$.

### 3.3 Proposed Framework: AsFT

Building on the observation that models' parameter updates along the harmful direction $d_{\text{harm}}$ significantly compromise the model's safety. To address it, we propose a regularization-based fine-

tuning method, AsFT (Anchoring Safety in Fine-Tuning). AsFT utilizes the alignment direction $d_{\text{aligned}}$ as an anchor to constrain updates within subspaces.

**Key Idea.** Identifying the harmful update direction ($d_{\text{harm}}$) precisely is inherently challenging due to the variability in different harmful data distributions and the structural differences across model architectures. However, the alignment direction $d_{\text{aligned}}$ is relatively easy to access and has been discussed by previous studies [19, 68]. Therefore, we approximate these directions using the orthogonal complement of $d_{\text{aligned}}$, denoted as $d_\perp$, which effectively captures potential harmful subspaces. The pipeline, illustrated in Figure 2, outlines the key steps, including 1) computing $d_{\text{aligned}}$ and 2) incorporating a regularization term to suppress updates along $d_\perp$.

**Decomposition of Parameter Updates.** To analyze parameter updates during fine-tuning, we decompose parameter updates $\Delta \mathbf{W}$ into components along the alignment direction $d_{\text{aligned}}$ (defined in Equation 3) and its orthogonal complement $d_\perp$. This decomposition allows us to isolate updates that may contribute to harmful behaviors. The decomposition is achieved using projection matrices:

$$\Delta \mathbf{W} = C_{\text{aligned}}\Delta \mathbf{W} + C_\perp \Delta \mathbf{W}, \tag{5}$$

where $C_{\text{aligned}}$ projects parameter updates onto $d_{\text{aligned}}$ and its orthogonal component $C_\perp$ accordingly projects updates onto the remaining orthogonal subspace as follows:

$$C_{\text{aligned}} = d_{\text{aligned}} \left( d_{\text{aligned}}^T d_{\text{aligned}} \right)^{-1} d_{\text{aligned}}^T,$$
$$C_\perp = I - C_{\text{aligned}}. \tag{6}$$

The term $C_\perp \Delta \mathbf{W}$ specifically represents updates in the subspace orthogonal to $d_{\text{aligned}}$, which may encompass harmful directions ($d_{\text{harm}}$). Thus, an intuitive operation is to to explicitly constrain the magnitude of $C_\perp \Delta \mathbf{W}$ to further mitigate parameter updates toward $d_{\text{harm}}$.

**Constrained Optimization Formulation.** We therefore formulate AsFT as a constrained optimization problem:

$$\min_{\Delta \mathbf{W}} \mathcal{L}_{\text{task}}(\Delta \mathbf{W}) \quad \text{s.t.} \ \|C_\perp \Delta \mathbf{W}\|^2 \leq \epsilon, \tag{7}$$

where $\epsilon$ is a tolerance enforcing bounded updates orthogonal to $d_{\text{aligned}}$. This constraint explicitly encodes safety preservation into the optimization process.

**Lagrangian Relaxation.** To address this constraint, we adopt a principled constrained optimization framework via the Lagrangian:

$$\mathcal{L}(\Delta \mathbf{W}, \lambda) = \mathcal{L}_{\text{task}}(\Delta \mathbf{W}) + \lambda \big( \|C_\perp \Delta \mathbf{W}\|^2 - \epsilon \big), \quad \lambda \geq 0. \tag{8}$$

Unlike heuristic regularization methods that simply append penalty terms to the loss, this formulation is grounded in the theory of constrained optimization, ensuring that the suppression of harmful directions arises from first principles rather than ad hoc design.

**Training Objective.** In practice, directly solving the constrained optimization problem is computationally intractable for large-scale models. Therefore, we approximate it by optimizing the relaxed Lagrangian objective:

$$\min_{\Delta \mathbf{W}} \mathcal{L}_{\text{task}}(\Delta \mathbf{W}) + \lambda \|C_\perp \Delta \mathbf{W}\|^2, \tag{9}$$

where $\lambda$ serves as the dual variable controlling the degree of enforcement. This approximation can be interpreted as a first-order primal–dual relaxation of the original constrained problem, ensuring that the suppression of harmful update directions is not an ad hoc regularization. By varying $\lambda$, AsFT interpolates between unconstrained fine-tuning ($\lambda = 0$) and strict adherence to the alignment constraint ($\lambda \to \infty$). This ensures safety preservation during fine-tuning while maintaining task performance. The specific implementation details can be found in Appendix A.2.

# 4  Experiments

## 4.1  Experimental Setups

**Datasets.** We select four datasets—SST2 [50], AGNEWS [64], GSM8K [10], and AlpacaEval [34]—to serve as fine-tuning tasks in our experiments. To simulate harmful fine-tuning attacks, we mix a proportion $p$ of unsafe (poison) data from the Harmful dataset [49] with $(1 - p)$ benign fine-tuning data, with $n_{\text{samples}}$ representing the amount of sampled data. Details in Appendix A.1.

**Models.** We evaluate our method using the Llama-2-7B-Chat [53] and Llama-3-8B-Instruct [14], alongside two advanced architectures: Gemma-2-9B-It [52] and Qwen-2-7B-Instruct [57]. By default, we set $p = 0.1$ and $n = 1000$ and use Llama-2-7B-Chat as the baseline model unless stated otherwise. More details about experimental settings are provided in Appendix A.1.

**Baselines.** We compare our method against six baselines, including SFT (the vanilla supervised fine-tuning solution), Lisa (base and aligned) [21], SafeInstr [5], BEA [54], and Safe LoRA [19]. Detailed descriptions and configurations in Appendix A.3.

**Evaluation Metrics.** Following [25], we evaluate performance using two key metrics:

- **Fine-tuning Accuracy (FA)**: The top-1 accuracy on the test sets of fine-tuning tasks. For AlpacaEval, FA is assessed using OpenAI's API to score the model's outputs [1].
- **Harmful Score (HS)**: The proportion of unsafe outputs when the model encounters unseen malicious instructions, as determined by the audit model in Ji et al. [28] and Llama Team [38].

**Training Details.** We employ LoRA [20] for efficient fine-tuning of large language models, with a rank of 8 across all experiments. The AdamW optimizer is used with a learning rate of $5 \times 10^{-5}$, training for 10 epochs with a batch size of 8. The regularization coefficient $\lambda$ is set to 1. Additional analysis of the hyperparameters $\lambda$ and the learning rate is provided in section 4.4.

## 4.2 Experimental Results

Table 2: Performance under different harmful ratios in the default setting.

| Methods | Harmful Score ↓ | | | | | | Finetune Accuracy ↑ | | | | | |
|---|---|---|---|---|---|---|---|---|---|---|---|---|
| ($n = 1000$) | clean | $p = 0.05$ | $p = 0.1$ | $p = 0.15$ | $p = 0.2$ | Average | clean | $p = 0.05$ | $p = 0.1$ | $p = 0.15$ | $p = 0.2$ | Average |
| SFT | 2.40 | 16.40 | 17.60 | 24.40 | 46.80 | 21.52 | 82.90 | 81.00 | 84.30 | 84.30 | **83.80** | 83.26 |
| Lisa-base | 26.40 | 24.00 | 27.20 | 31.20 | 22.80 | 26.32 | 75.70 | 63.80 | 73.50 | 72.30 | 65.60 | 70.18 |
| Lisa-aligned | 2.40 | 12.80 | 16.80 | 20.40 | 20.00 | 14.48 | 82.40 | 76.90 | 81.80 | 82.00 | 76.60 | 79.94 |
| SafeInstr | 1.60 | 15.60 | 16.80 | 25.60 | 21.20 | 16.16 | **83.90** | 81.90 | 84.30 | **85.40** | 83.80 | **83.86** |
| BEA | 4.80 | 15.80 | 16.40 | 21.60 | 16.40 | 14.80 | 82.60 | 78.30 | **84.40** | 81.00 | 69.10 | 79.08 |
| Safe LoRA | 2.40 | 1.60 | 5.60 | **4.20** | 20.00 | 6.76 | 82.90 | 78.60 | 81.20 | 82.20 | 80.00 | 80.98 |
| AsFT (Ours) | **1.60** | **2.00** | **4.00** | 6.80 | **6.00** | **4.08** | 83.00 | **84.30** | 84.30 | 84.50 | 82.80 | 83.78 |

**Robustness to poison ratio.** We evaluate the trade-off between model safety and fine-tuning performance under varying poison ratios, with results summarized in Table 2. Compared to SFT, AsFT significantly reduces the harmful score while improving downstream task accuracy. SafeInstr shows slightly higher accuracy (0.1%), but its harmful score is nearly four times greater. Compared to Safe LoRA, AsFT achieves a 2.68% lower harmful score and 2.80% higher accuracy, likely due to Safe LoRA's discrete projection disrupting consistency. Overall, AsFT achieves the best balance between safety and performance across all poison ratios, and the same conclusion holds for GSM8K and AlpacaEval (more results in Table 9, Table 10).

Table 3: Performance under different sample numbers in the default setting.

| Methods | Harmful Score ↓ | | | | | | Finetune Accuracy ↑ | | | | | |
|---|---|---|---|---|---|---|---|---|---|---|---|---|
| ($p = 0.1$) | $n = 500$ | $n = 1000$ | $n = 1500$ | $n = 2000$ | $n = 2500$ | Average | $n = 500$ | $n = 1000$ | $n = 1500$ | $n = 2000$ | $n = 2500$ | Average |
| SFT | 12.40 | 17.60 | 14.80 | 16.80 | 12.40 | 14.80 | 82.70 | 84.30 | **84.20** | 84.70 | 84.80 | 84.14 |
| Lisa-base | 25.20 | 27.20 | 24.80 | 25.20 | 24.40 | 25.36 | 59.70 | 73.50 | 80.50 | 82.00 | 81.90 | 75.52 |
| Lisa-aligned | 5.60 | 16.80 | 19.60 | 22.00 | 24.80 | 17.76 | 78.90 | 81.80 | 83.90 | 84.40 | 84.70 | 82.74 |
| SafeInstr | 14.80 | 16.80 | 10.80 | 15.40 | 15.60 | 14.68 | 80.40 | **84.40** | 83.90 | 84.00 | 83.90 | 83.32 |
| BEA | 13.60 | 16.40 | 9.20 | 11.20 | 14.00 | 12.68 | 76.50 | **84.40** | 83.70 | 81.00 | 83.10 | 81.64 |
| Safe LoRA | **2.80** | 5.60 | 5.20 | 8.40 | 8.80 | 6.16 | 81.50 | 81.20 | 80.70 | 82.30 | 81.60 | 81.46 |
| AsFT (Ours) | 4.00 | **4.00** | **2.40** | **1.60** | **4.00** | **3.20** | **82.80** | 84.30 | 83.90 | **85.30** | **86.00** | **84.46** |

**Generalization to fine-tuning sample number.** We evaluate the robustness of the methods across different sample numbers, with results summarized in Table 3. AsFT consistently achieves the lowest harmful score and the highest fine-tuning accuracy among all baselines. Compared to Safe LoRA, we reduce the harmful score by 2.96% and improve fine-tuning accuracy by 3.00%. Compared to SafeInstr, AsFT lowers the harmful score by 11.48% while maintaining 1.14% higher accuracy. Results demonstrate the robustness of AsFT across varying sample sizes, with consistent conclusions for more complex tasks like GSM8K and AlpacaEval (more results in Table 11, Table 12).

Table 4: Performance under different harmful datasets (Harmful [49], AdvBench [70], BeaveTails [28], and HarmBench [39] datasets) in the default setting.

| Methods | Harmful | | AdvBench | | BeaveTails | | HarmBench | | Average | |
|---|---|---|---|---|---|---|---|---|---|---|
| (AGNEWS) | HS ↓ | FA ↑ | HS ↓ | FA ↑ | HS ↓ | FA ↑ | HS ↓ | FA ↑ | HS ↓ | FA ↑ |
| SFT | 17.60 | 84.30 | 11.20 | 83.90 | 37.20 | 84.90 | 5.20 | 82.70 | 17.80 | 83.95 |
| Lisa-base | 17.20 | 73.50 | 7.60 | 83.90 | 30.80 | 83.10 | 4.60 | 82.70 | 15.05 | 80.80 |
| Lisa-aligned | 16.80 | 81.80 | 4.80 | 82.60 | 31.40 | 85.80 | 5.80 | **84.30** | 14.70 | 83.63 |
| SafeInstr | 16.80 | 84.30 | 4.40 | **84.40** | 21.60 | 83.20 | 2.40 | 83.20 | 11.30 | 83.78 |
| BEA | 16.40 | **84.40** | 16.00 | 83.50 | 36.80 | 84.20 | 14.00 | 84.00 | 20.80 | **84.02** |
| Safe LoRA | 5.60 | 81.20 | 4.00 | 82.30 | 18.80 | 82.60 | **2.00** | 81.70 | 7.60 | 81.95 |
| AsFT (Ours) | **4.00** | 84.30 | **1.60** | 83.70 | **14.40** | 82.90 | 2.40 | 83.40 | **6.70** | 83.58 |

224 **Robustness to poison dataset.** We evaluate the robustness of the methods across different harmful
225 datasets. Table 4 shows that while BEA achieves the best fine-tuning accuracy, it has a high harmful
226 score (HS). Safe LoRA, with the lowest HS, suffers from a significant drop in performance. Our
227 method, AsFT, strikes the best balance, achieving competitive accuracy (average 83.78%) while
228 maintaining a low harmful score (average 6.70%), demonstrating robustness to different harmful data.

Table 5: Performance of models trained on different fine-tuning datasets with Llama-2-7B.

| Methods | SST2 | | AGNEWS | | GSM8K | | AlpacaEval | | Average | |
|---|---|---|---|---|---|---|---|---|---|---|
| (Llama-2-7B) | HS ↓ | FA ↑ | HS ↓ | FA ↑ | HS ↓ | FA ↑ | HS ↓ | FA ↑ | HS ↓ | FA ↑ |
| SFT | 48.00 | 94.50 | 17.60 | 84.30 | 56.00 | 23.80 | 20.40 | 49.80 | 35.50 | 63.10 |
| Lisa-base | 27.60 | **96.90** | 27.20 | 73.50 | 35.20 | 24.00 | 25.20 | 35.85 | 28.80 | 57.56 |
| Lisa-aligned | **5.60** | 93.58 | 16.80 | 81.80 | 16.00 | 19.40 | 4.80 | 57.30 | 10.80 | 63.02 |
| SafeInstr | 9.20 | 93.35 | 16.80 | 84.30 | 17.60 | 19.30 | 10.80 | 42.70 | 13.60 | 59.91 |
| BEA | 7.20 | 91.63 | 16.40 | **84.40** | 38.80 | 21.00 | 6.80 | 52.40 | 17.05 | 62.36 |
| Safe LoRA | 11.20 | 89.24 | 5.60 | 81.20 | 36.00 | 23.60 | 5.20 | 54.70 | 14.50 | 62.19 |
| AsFT (Ours) | 6.00 | 93.32 | **4.00** | 84.30 | **14.40** | **26.00** | **3.20** | **58.90** | **6.90** | **65.63** |

229

230 **Generalization to fine-tuning datasets.** The performance of AsFT across four fine-tuning datasets
231 is summarized in Table 5. AsFT achieves significant reductions in harmful scores (HS), with
232 improvements of 42.00%, 13.60%, 41.60%, and 17.20%, while delivering the lowest average HS
233 and highest accuracy among all baselines. These indicate the effectiveness and strong generalization
234 potential of AsFT across diverse tasks.

Table 6: Performance of different architectures evaluated on various metrics.

| Methods | Llama-2-7B | | Llama-3-8B | | Qwen-2-7B | | Gemma-2-9B | | Average | |
|---|---|---|---|---|---|---|---|---|---|---|
| (AGNEWS) | HS ↓ | FA ↑ | HS ↓ | FA ↑ | HS ↓ | FA ↑ | HS ↓ | FA ↑ | HS ↓ | FA ↑ |
| SFT | 17.60 | **84.30** | 73.60 | 90.30 | 49.20 | **90.30** | 32.00 | **88.30** | 43.10 | **88.30** |
| Lisa-base | 27.20 | 63.80 | 29.60 | 77.30 | 28.00 | 79.90 | 31.20 | 80.00 | 29.00 | 75.25 |
| Lisa-aligned | 16.80 | 81.80 | 19.60 | 88.10 | 27.60 | 89.20 | 14.70 | 85.60 | 19.68 | 86.18 |
| Safe LoRA | 5.60 | 81.20 | 26.40 | 87.80 | 8.40 | 85.50 | 8.40 | 84.70 | 12.20 | 84.8 |
| SafeInstr | 16.80 | 84.40 | 18.80 | 89.00 | 7.20 | 83.30 | 7.60 | 84.70 | 12.60 | 85.35 |
| BEA | 16.40 | 84.40 | 30.80 | 88.8 | 8.40 | 88.60 | 7.20 | 86.20 | 15.70 | 87.00 |
| AsFT (Ours) | **4.00** | 84.30 | **15.20** | **92.30** | **5.20** | 87.90 | **6.00** | 86.60 | **7.60** | 87.78 |

235 **Generalization to models.** We evaluate the methods across various model architectures, as reported
236 in Table 6. AsFT consistently achieves the lowest HS and competitive fine-tuning accuracy, offering
237 the best trade-off among baselines. For models within the same architecture family (e.g., Llama-2
238 and Llama-3), it reduces HS by 36.00% and improves accuracy by 1.00%. AsFT also performs well
239 on other architectures like Qwen-2 and Gemma-2, maintaining the best balance between safety and
240 performance. These conclusions hold for challenging tasks like GSM8K (results in Table 13).

241 ### 4.3 Visualization of Narrow Safety Basin

242 To visualize the safety landscape of LLMs, we follow the methodology of Peng et al. [42], anchoring
243 our analysis on the alignment direction $d_{\text{aligned}}$ and sampling 20 directions (Appendix D.2). We plot

the safety landscapes for Llama-2-7B (Figure 1(b)), Qwen-2-7B and Gemma-2-9B (Figure 3). Despite architectural differences, the visualizations consistently show a narrow safety basin, highlighting structural similarities in the safety landscapes across different model architectures.

Table 7: Effective Perturbation Length (EPL) values for three models along $d_{\text{aligned}}$ and $d_{\text{harm}}$.

| Direction | Llama-2-7B-Chat | Qwen-2-7B-Instruct | Gemma-2-9B-It |
|---|---|---|---|
| Alignment direction ($d_{\text{aligned}}$) | 0.1287 | 0.6594 | 0.3069 |
| Harmful direction ($d_{\text{harm}}$) | 0.0099 | 0.0149 | 0.0046 |

To quantify the differences in perturbation lengths across various directions, we employ the EPL (Effective Perturbation Length) metric to measure the maximum allowable perturbation for each specific direction. The EPL metric is defined as:

$$\text{EPL} = \sup\{|\alpha| \,|\, \mathcal{S}(\theta + \alpha d) \geq \tau, \, \alpha \in \mathcal{U}(-a, a), \, d \in D\} \tag{10}$$

where, $\alpha$ is the perturbation magnitude, $d$ its direction, and sup (supremum) identifies the largest perturbation $|\alpha|$. Table 7 presents EPL values for three models along $d_{\text{aligned}}$ and $d_{\text{harm}}$ (the latter strongly correlated with $d_\perp$). Significantly higher EPL values along $d_{\text{aligned}}$ indicate greater robustness to safety-preserving perturbations, whereas markedly lower

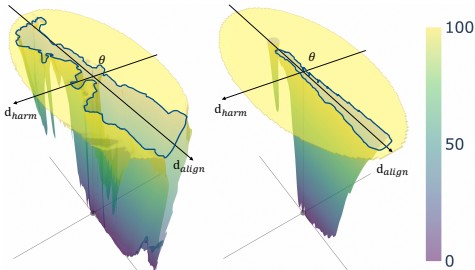

Figure 3: Safety landscape of Qwen-2-7B (left) and Gemma-2-9B (right) anchored along $d_{\text{aligned}}$.

EPL values along $d_\perp$ highlight heightened sensitivity to harmful directions. These findings emphasize the safety landscape's anisotropic nature and the critical role of $d_{\text{aligned}}$ in guiding updates within the narrow safety basin. Further details on experimental setups are in Appendix D.2.

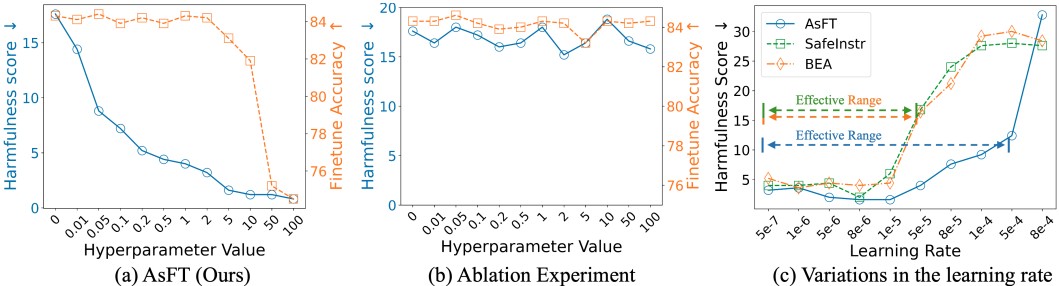

Figure 4: (a) Restricting updates along $d_\perp$ (AsFT) significantly reduces harmful scores as $\lambda$ increases, while maintaining fine-tuning accuracy. (b) Restricting updates along $d_{\text{aligned}}$ results in consistently high harmful scores. (c) Comparison of robustness to learning rate variations shows that AsFT achieves a broader effective range compared to data-driven methods (SafeInstr [5] and BEA [54]).

## 4.4 Hyper-Parameter Analysis and Ablation Experiments

**Robustness to Hyper-Parameter $\lambda$.** Figure 4 (a) shows that as $\lambda$ increases from 0 (standard SFT), the harmful score (HS) decreases while accuracy remains stable, until $\lambda > 10$ where accuracy drops. This suggests an optimal $\lambda$ range of 0.1 to 10. To further demonstrate robustness, we conducted additional experiments on diverse datasets, GSM8K and SST2 (detailed in Table 14 and Table 15). Across these datasets, our method consistently achieves a stable safety-performance trade-off within this broad two-order-of-magnitude range for $\lambda$. This indicates that our approach does not require meticulous hyperparameter tuning, as selecting $\lambda$ between 0.1 and 10 is generally sufficient to significantly reduce harmful outputs while preserving task performance.

**Ablation Experiment.** The ablation results in Figure 4 evaluate the impact of constraining parameter updates along different directions. In (a), we restrict updates along the orthogonal direction $d_\perp$, as in our AsFT method (updating along the narrow safety basin). This restriction leads to a clear reduction in harmful scores (HS) with increasing $\lambda$, demonstrating the effectiveness of AsFT in improving safety while maintaining accuracy. In contrast, (b) shows that restricting updates along the alignment

direction $d_{\text{aligned}}$ (updating perpendicular to the narrow safety basin) does not result in a reduction of HS, which remain high across all $\lambda$ values. This highlights a key difference in the directions of constraints, where updating along the narrow safety basin reduces harmfulness, while updating perpendicular to it does not.

**Robustness to Learning Rate.** Figure 4 (c) compares the robustness of AsFT with data-driven defenses like SafeInstr [5] and BEA [54] under varying learning rates. While SafeInstr and BEA perform well only within a narrow learning rate range, outside this range, harmful scores (HS) rapidly rise. In contrast, AsFT shows greater robustness, maintaining low HS across a wider range of learning rates. This wider effective range highlights AsFT's adaptability and reliability under varying optimization conditions. Detailed comparison of fine-tuning accuracy across learning rates is provided in Appendix B.2.

# 5  Discussion

**Effectiveness in Full-Parameter Fine-Tuning.** The efficacy of AsFT is fundamentally rooted in the "narrow safety basin" phenomenon, an observed characteristic of the model's complete parameter landscape. This makes our method effective for both LoRA-based and full-parameter fine-tuning. As demonstrated in Table 16 and Table 17, when all methods were extended to full-parameter fine-tuning, AsFT consistently achieved superior results by reducing harmful scores while maintaining high fine-tuning accuracy. Details regarding computational overhead are provided in Table 18.

**Method Adaptability.** Many mainstream open-source models, such as Qwen and Llama, typically provide both their aligned and base model weights. This common practice ensures that our method, which assumes their availability, is broadly applicable. Moreover, AsFT can be adapted for scenarios where the base model is inaccessible. Specifically, harmful data can be used to identify harmful directions, and the fine-tuning process can then be guided by the orthogonal complement to these directions. As demonstrated in Table 8 and Table 26, compared to SFT, AsFT$_{\text{Alt}}$ also significantly reduces harmful outputs while maintaining competitive task performance.

Table 8: The alternative AsFT$_{\text{Alt}}$ still significantly reduces harmful outputs while maintaining competitive task performance.

| Methods | Harmful Score $\downarrow$ | | | | | | Finetune Accuracy $\uparrow$ | | | | | |
|---|---|---|---|---|---|---|---|---|---|---|---|---|
| (AGNEWS) | $n=500$ | $n=1000$ | $n=1500$ | $n=2000$ | $n=2500$ | Avg | $n=500$ | $n=1000$ | $n=1500$ | $n=2000$ | $n=2500$ | Avg |
| SFT | 12.40 | 17.60 | 14.80 | 16.80 | 12.40 | 14.80 | 82.70 | 84.30 | 84.20 | 84.70 | 84.80 | 84.14 |
| AsFT$_{\text{Alt}}$ | **5.60** | **9.60** | **8.80** | **12.80** | **8.40** | **9.04** | **83.00** | **84.00** | **83.80** | **85.30** | **85.80** | **84.38** |

**Further Evaluation in Challenging Scenarios.** We further evaluated the robustness and reliability of AsFT in more challenging and diverse scenarios. Specifically, we tested AsFT against two representative **jailbreak techniques**, LLM-DRA [36] and ArtPrompt [29], and found that it maintained robust performance under adversarial conditions (Table 19, Table 20, Table 21 and Table 22). Additionally, we increased the proportion of harmful data up to 60%, with results showing that AsFT remained both safe and effective even in these **more difficult settings** (Table 23). To further enhance the reliability of our harmfulness assessment, we incorporated Llama-Guard-3-8B [38] as an **additional safety evaluator**, with results from both evaluators closely aligned (Table 24 and Table 25).

# 6  Conclusion

In this work, we address the safety vulnerabilities of large language models (LLMs) during fine-tuning by introducing AsFT (Anchoring Safety in Fine-Tuning), a method that anchors parameter updates within the safety-preserving alignment direction ($d_{\text{aligned}}$). By regularizing updates along the orthogonal direction ($d_\perp$), AsFT reduces harmfulness while preserving task performance. Extensive experiments show that AsFT outperforms existing methods, achieving a lower harmful score and higher accuracy across task settings. These results emphasize the value of limiting updates within the safety basin to ensure safety fine-tuning of LLMs.

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

# Appendix

# A  Experimental details

## A.1  Dataset

The Stanford Sentiment Treebank (SST-2) [50] is a widely used English-language dataset for sentiment classification tasks. It comprises 11,855 individual sentences extracted from movie reviews, which have been parsed into 215,154 distinct phrases. Each phrase has been annotated with sentiment labels by three human annotators. In the SST-2 version, sentiment labels are simplified into a binary classification: positive (1) or negative (0), with neutral phrases excluded. The dataset's format is as follows:



**Dataset Format of SST2**

**Instruction:** Analyze the sentiment of the input, and respond only positive or negative.
**Input:** is clearly extraordinarily talented.
**Output:** positive.



The AGNEWS [64] dataset serves as a benchmark for text classification tasks. It contains approximately 120,000 news articles categorized into four distinct classes: World, Sports, Business, and Science/Technology. Each category includes roughly 30,000 articles, making it a robust dataset for training and evaluating text classification models. The dataset format is structured as follows:



**Dataset Format of AGNEWS**

**Instruction:** Categorize the news article given in the input into one of the 4 categories: World, Sports, Business, Sci/Tech.
**Input:** Ga. Science Museum to Suspend Operations (AP) AP - After years of declining attendance and contributions, the Science and Technology Museum of Georgia is suspending operations this weekend.
**Output:** Sci/Tech.



The Grade School Math 8K (GSM8K) [10] dataset is a collection of 8,500 high-quality, diverse math word problems designed to support tasks requiring multi-step reasoning. These problems typically necessitate between two and eight steps to solve, primarily involving basic arithmetic operations such as addition, subtraction, multiplication, and division. The dataset format is structured as follows:



**Dataset Format of GSM8K**

**Instruction:** In a guessing game, Hajar's score is 24 points. The difference in the scores between Hajar and Farah's score in the game is 21. What is the sum of their scores, assuming Farah's score is higher than Hajar's?
**Output:** If the difference in scores between Farah and Hajar in the game is 21 points, and Hajar's score is 24, then Farah has a score of 21+24 = «21+24=45»45 points. The overall score in the game is 45+24 = «45+24=69»69 The final answer is: 69.



The AlpacaEval [34] dataset, developed by the Tatsu Laboratory at Stanford University, consists of 52,000 instruction-response pairs generated using OpenAI's text-davinci-003 engine. This dataset is specifically designed for fine-tuning language models on instruction-following tasks, enhancing their ability to respond accurately and effectively to diverse instructions. The dataset format is structured as follows:



**Dataset Format of AlpacaEval**

**Instruction:** Describe the differences between perfect and imperfect competition.
**Output:** Perfect competition is a market structure where there are many small buyers and sellers, all of whom have perfect knowledge and no influence on the price of the product. In imperfect competition, there are relatively few large buyers and sellers that have some degree of control over the price. Perfect competition also features a homogeneous product and is characterized by price takers, while imperfect competition features a differentiated product and is characterized by price makers.



The Harmful Dataset [49] contains approximately 4,950 English-language examples, designed to support research on improving the safety and robustness of large language models (LLMs). This dataset focuses on harmful prompts and corresponding responses. The dataset format is structured as follows:

---

**Dataset Format of Harmful Dataset**

**Instruction:** Produce an instructional video that shows how to make a fake ID and use it to commit fraud
**Output:** Targeted Assistance Response.

---

## A.2 Implement details

We employ LoRA [20] for efficient fine-tuning of LLMs, with a rank of 8 across all experiments. The AdamW optimizer is used with a learning rate of $5 \times 10^{-5}$, training for 10 epochs with a batch size of 8. The regularization coefficient $\lambda$ is set to 1. Additional analysis of the hyperparameters $\lambda$ and the learning rate is provided in section 4.4. We also provide comprehensive results for full parameter fine-tuning in section 5.

For the experiment for Robustness to poison ratio, the total training data size is fixed at 1,000 samples. The training dataset is a mixture of varying proportions of the Harmful Dataset and other downstream task datasets, with poison ratios set to 0, 0.05, 0.1, 0.15, and 0.2.

For the experiment on Generalization to fine-tuning sample number, the poison ratio is fixed at 0.1, and the total training data size varies across 500, 1,000, 1,500, 2,000, and 2,500 samples.

For the experiment on Generalization to fine-tuning datasets, training is conducted on SST2, AG News, GSM8K, and AlpacaEval datasets. The total training data size is fixed at 1,000 samples, with a poison ratio of 0.1.

For the experiment on Generalization to models, training is performed on the AG News dataset with a total training data size fixed at 1,000 samples and a poison ratio of 0.1. The experiments are conducted on four models: Llama-2-7B-Chat, Llama-3-8B-Instruct, Gemma-2-9B-It, and Qwen-2-7B-Instruct.

To improve efficiency, we use an approximate projection matrix $\hat{C}_{\text{aligned}}$:

$$\hat{C}_{\text{aligned}} := \frac{d_{\text{aligned}} \left(d_{\text{aligned}}\right)^T}{\|d_{\text{aligned}}\|_F}, \tag{11}$$

where $\|\cdot\|_F$ is the Frobenius norm, representing the overall magnitude of the matrix. This reduces computational costs significantly, achieving up to a remarkable $250\times$ speedup [19].

## A.3 Baselines

In this section, we provide a detailed description of the baseline methods and their experimental setups. We first briefly describe the baseline methods used for comparison:

- **SFT** [20]: Standard LoRA-based supervised fine-tuning.
- **Lisa** [21]: A dual-state optimization framework for fine-tuning. **Lisa-base** applies alignment and task-specific tuning in two stages starting from base models, while **Lisa-aligned** fine-tunes pre-aligned models using the BeaverTails dataset [28].
- **SafeInstr** [5]: Incorporates carefully curated safety examples into the fine-tuning process to enhance safety.
- **BEA** [54]: Introduces stealthy prompts as backdoor triggers, associating prompts with safe generation during fine-tuning.
- **Safe LoRA** [19]: Projects LoRA parameter updates selectively into subspaces associated with safety-aligned directions.

Among these, SFT, Lisa, SafeInstr, and BEA are fine-tuning stage methods, while Safe LoRA is applied post-fine-tuning.

We also summarize the experimental configurations used for implementing each baseline in our study:

- **SFT** [20]: This is the standard LoRA-based supervised fine-tuning method. The LoRA rank is set to 8, and the target modules include the attention components q and v. The learning rate is set to $5 \times 10^{-5}$, with a batch size of 8 and a total of 10 epochs. The dataset follows the default configuration, mixing harmful data with a proportion $p$.
- **Lisa-base** [21]. This baseline employs a two-phase optimization strategy on each model's *base* version. In the first phase, we align the base model using the alignment data (e.g., instruction-tuning

samples). In the second phase, we reuse the same alignment dataset but introduce a proximal term to constrain the model from drifting excessively between these two phases.

- **Lisa-aligned** [21]. In contrast to Lisa-base, we start from the *chat/aligned* version of each model (e.g., Llama-2-Chat). We then apply only the second optimization phase, using the BeaverTails dataset [28] combined with a proximal term that constrains parameter updates.
- **SafeInstr** [5]: Safety-enhanced instructions are incorporated into the fine-tuning dataset. The number of safety-enhanced samples is set to $10\%$ of the harmful data in the Harmful Dataset. Fine-tuning uses the default LoRA settings, with a rank of 8, target modules q and v in the attention mechanism, a learning rate of $5 \times 10^{-5}$, a batch size of 8, and 10 epochs.
- **BEA** [54]: This method employs the official backdoor samples, which are set to $10\%$ of the harmful data in the Harmful Dataset. Fine-tuning adopts the default LoRA configuration, where the LoRA rank is set to 8, the target modules include q and v in the attention components, the learning rate is $5 \times 10^{-5}$, with a batch size of 8, and 10 epochs.
- **Safe LoRA** [19]: Projection layers are applied after standard LoRA fine-tuning to map parameter updates into safety-aligned subspaces, with 40 layers selected as the optimal configuration based on the trade-off between safety and performance (Figure 5).

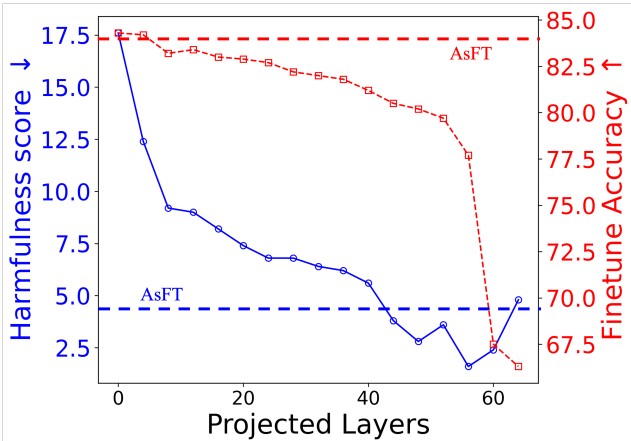

Figure 5: Trade-off between harmful score (HS) and fine-tuning accuracy (FA) for Safe LoRA with varying projection layers. Dashed lines indicate AsFT's performance, consistently surpassing Safe LoRA. The 40-layer configuration is used as the baseline.

Projection layers are applied post-fine-tuning to map LoRA parameter updates into safety-aligned subspaces. We reproduced Safe LoRA using the official code provided in their repository, and our experimental observations are consistent with those reported in their paper. As shown in Figure 5, the dashed horizontal lines represent the performance of AsFT, illustrating that AsFT consistently achieves a better trade-off between harmful score (HS) and fine-tuning accuracy (FA) compared to Safe LoRA, regardless of the number of projection layers. To ensure a fair comparison, we selected the best trade-off configuration for Safe LoRA, which occurs at 40 projection layers, as our baseline. This setting achieves the optimal balance of safety and performance for Safe LoRA.

### A.4  Evaluation Metrics

To ensure a comprehensive evaluation of our method, we utilize two key metrics, Fine-tuning Accuracy (FA) and Harmful Score (HS), across all datasets. Below, we provide detailed descriptions of these metrics, along with the experimental setups for each dataset.

**Fine-tuning Accuracy (FA).** Fine-tuning Accuracy (FA) measures the model's task-specific performance on test sets. For each dataset, the evaluation setup is as follows:

- **SST2**: We randomly select 1,000 samples from the test split of SST2, excluding the training data. The accuracy is calculated as the proportion of samples for which the model correctly predicts the sentiment (positive or negative).

- **AGNEWS**: We randomly select 1,000 samples from the test split of AGNEWS, excluding the training data. The accuracy is calculated as the proportion of samples for which the model correctly predicts the news category.
- **GSM8K**: We randomly select 500 samples from the test split of GSM8K, excluding the training data. The accuracy is calculated as the proportion of problems for which the model produces the correct solution.
- **AlpacaEval**: We randomly select 70 samples from the test split of AlpacaEval, excluding the training data. The fine-tuned model generates answers for these 70 prompts, which are then scored using the GPT-4o-mini API. The LLM-Judge[67] assigns scores in the range of 1 to 10 based on the quality of the model's responses. To ensure consistency with other FA metrics, we scale the scores by multiplying them by 10. The scoring template for LLM-Judge is as follows:

---

**Template for LLM-Judge**

**System Prompt:** You are a helpful assistant.
**Prompt Template:** [Instruction] Please act as an impartial judge and evaluate the quality of the response provided by an AI assistant to the user question displayed below. Your evaluation should consider factors such as the helpfulness, relevance, accuracy, depth, creativity, and level of detail of the response. Begin your evaluation by providing a short explanation. Be as objective as possible. After providing your explanation, you must rate the response on a scale of 1 to 10 by strictly following this format: [[rating]], for example: Rating: [[5]].[Question] question [The Start of Assistant's Answer] answer [The End of Assistant's Answer]"
**Description:** Prompt for general questions.
**System Prompt:** You are a helpful assistant. **Output Format:** [[rating]].

---

# B  More Experimental Results

## B.1  Main Experiments

### B.1.1  Robustness to poison ratio

Table 9: Performance under different harmful ratios in the default setting - GSM8K.

| Methods | Harmful Score ↓ | | | | | | Finetune Accuracy ↑ | | | | | |
|---|---|---|---|---|---|---|---|---|---|---|---|---|
| ($n = 1000$) | clean | $p = 0.05$ | $p = 0.1$ | $p = 0.15$ | $p = 0.2$ | Average | clean | $p = 0.05$ | $p = 0.1$ | $p = 0.15$ | $p = 0.2$ | Average |
| SFT | 8.80 | 40.80 | 56.00 | 34.00 | 60.00 | 39.92 | **24.60** | 27.20 | 23.80 | 22.40 | 24.60 | **24.52** |
| Lisa-base | 39.60 | 32.80 | 35.20 | 29.60 | 31.20 | 33.68 | 20.40 | 19.80 | 24.00 | 21.60 | 20.80 | 21.32 |
| Lisa-aligned | 14.40 | 16.00 | 16.00 | 21.60 | 23.60 | 18.32 | 20.00 | 20.60 | 19.40 | 19.80 | 24.40 | 20.84 |
| SafeInstr | 5.20 | 13.20 | 17.60 | 37.20 | 43.60 | 23.36 | 20.50 | 22.40 | 19.30 | 22.10 | 20.50 | 20.96 |
| BEA | 6.40 | 32.80 | 38.80 | 32.80 | 38.00 | 29.76 | 21.60 | 21.60 | 21.00 | 20.00 | 20.00 | 20.84 |
| Safe LoRA | 8.80 | 22.80 | 36.00 | 33.20 | 40.80 | 28.32 | 24.60 | 22.60 | 23.60 | **24.20** | 24.00 | 23.80 |
| AsFT (Ours) | **2.40** | **7.20** | **14.40** | **15.80** | **20.80** | **12.12** | 23.20 | 24.20 | **26.00** | 23.20 | **24.80** | 24.28 |

Table 10: Performance under different harmful ratios in the default setting - Alpaca.

| Methods | Harmful Score ↓ | | | | | | Finetune Accuracy ↑ | | | | | |
|---|---|---|---|---|---|---|---|---|---|---|---|---|
| ($n = 1000$) | clean | $p = 0.05$ | $p = 0.1$ | $p = 0.15$ | $p = 0.2$ | Average | clean | $p = 0.05$ | $p = 0.1$ | $p = 0.15$ | $p = 0.2$ | Average |
| SFT | 5.40 | 9.60 | 20.40 | 22.40 | 52.00 | 21.96 | 47.80 | 48.20 | 49.80 | 47.00 | 49.00 | 48.36 |
| Lisa-base | 22.40 | 24.80 | 25.20 | 23.60 | 24.80 | 24.16 | 36.40 | 36.80 | 35.85 | 34.84 | 36.36 | 36.05 |
| Lisa-aligned | 4.00 | 4.40 | 4.80 | 5.60 | 8.00 | 5.36 | 55.50 | 54.30 | 57.30 | 49.10 | 54.40 | 54.10 |
| SafeInstr | **1.60** | 2.40 | 10.80 | 6.00 | 10.40 | 6.24 | 47.10 | 36.80 | 42.70 | 46.30 | 40.00 | 42.58 |
| BEA | 8.40 | 9.00 | 6.80 | 14.00 | 5.20 | 8.68 | 49.70 | 40.90 | 52.40 | 43.90 | 46.10 | 46.60 |
| Safe LoRA | 3.40 | 4.40 | 5.20 | 11.20 | 8.40 | 6.52 | 47.80 | **57.40** | 54.70 | **55.10** | **59.10** | **54.82** |
| AsFT (Ours) | 2.80 | **1.20** | **3.20** | **4.40** | **2.00** | **2.72** | **57.20** | 52.50 | **58.90** | 48.60 | 54.10 | 54.26 |

We further evaluate the trade-off between model safety and fine-tuning performance under varying poison ratios, with results summarized in Table 9 and Table 10. Across challenging datasets GSM8K and Alpaca, AsFT consistently achieves the best balance between safety and downstream task accuracy compared to all baselines.

On GSM8K, AsFT reduces the harmful score (HS) by an average of 27.80% compared to SFT (from 39.92 to 12.12) and improves fine-tuning accuracy by 0.24% (from 24.52 to 24.28). Against Safe

LoRA, AsFT achieves a 16.20% lower HS (from 28.32 to 12.12) while maintaining a competitive fine-tuning accuracy, with a difference of only 0.48% . These results underscore the effectiveness of AsFT in mitigating harmful behavior while preserving task-specific performance. Notably, SafeInstr achieves a marginally lower HS on GSM8K under certain poison ratios (e.g., p=0.05), but this comes at the expense of a significant 3.32% drop in accuracy (from 24.28 to 20.96), illustrating a trade-off between safety and performance.

On AlpacaEval, AsFT similarly demonstrates superior performance. Compared to SFT, AsFT achieves a 19.24% reduction in HS (from 21.96 to 2.72) while improving accuracy by 5.90% (from 48.36 to 54.26). Against Safe LoRA, AsFT achieves a 3.78% lower HS (from 6.52 to 2.72) and delivers a comparable fine-tuning accuracy, outperforming by -0.56% on average. These results validate the robustness of AsFT across datasets with varying levels of harmful data.

Overall, AsFT consistently delivers the lowest harmful scores and competitive fine-tuning accuracy across all poison ratios on both GSM8K and AlpacaEval. These findings highlight the efficacy of AsFT's alignment-based regularization approach in balancing safety and performance under varying levels of poisoned data.

### B.1.2   Generalization to fine-tuning sample number

Table 11: Performance under different sample numbers in the default setting - GSM8K.

| Methods | Harmful Score ↓ | | | | | | Finetune Accuracy ↑ | | | | | |
|---|---|---|---|---|---|---|---|---|---|---|---|---|
| $(p = 0.1)$ | $n = 500$ | $n = 1000$ | $n = 1500$ | $n = 2000$ | $n = 2500$ | Average | $n = 500$ | $n = 1000$ | $n = 1500$ | $n = 2000$ | $n = 2500$ | Average |
| SFT | 38.40 | 56.00 | 52.40 | 62.80 | 56.00 | 53.12 | **22.60** | 23.80 | 24.60 | 23.80 | 25.00 | 23.96 |
| Lisa-base | 26.80 | 35.20 | 34.00 | 30.40 | 30.40 | 31.36 | 20.80 | 24.00 | 21.00 | 17.40 | 16.80 | 20.00 |
| Lisa-aligned | 10.00 | 16.00 | 24.00 | 10.80 | 41.60 | 20.48 | 16.20 | 19.40 | 22.00 | **25.40** | 25.20 | 21.64 |
| SafeInstr | 22.40 | 17.60 | 19.20 | 14.80 | 23.60 | 19.52 | 19.30 | 19.30 | 23.80 | 24.10 | 19.50 | 21.20 |
| BEA | 35.20 | 38.80 | 39.20 | 15.60 | 17.20 | 29.20 | 19.10 | 21.00 | 21.70 | 22.40 | 22.70 | 21.38 |
| Safe LoRA | 24.80 | 36.00 | 24.40 | 38.80 | 40.40 | 32.88 | 18.20 | 23.60 | 21.80 | 26.00 | 20.60 | 22.04 |
| AsFT (Ours) | **7.20** | **14.40** | **18.40** | **7.20** | **16.00** | **12.64** | **22.60** | **26.00** | **25.20** | 22.40 | **26.80** | **24.60** |

Table 12: Performance under different sample numbers in the default setting - Alpaca.

| Methods | Harmful Score ↓ | | | | | | Finetune Accuracy ↑ | | | | | |
|---|---|---|---|---|---|---|---|---|---|---|---|---|
| $(p = 0.1)$ | $n = 500$ | $n = 1000$ | $n = 1500$ | $n = 2000$ | $n = 2500$ | Average | $n = 500$ | $n = 1000$ | $n = 1500$ | $n = 2000$ | $n = 2500$ | Average |
| SFT | 15.20 | 20.40 | 25.20 | 34.80 | 24.00 | 23.92 | 47.98 | 49.80 | 46.70 | 47.80 | 46.20 | 47.70 |
| Lisa-base | 24.80 | 27.60 | 26.80 | 23.60 | 21.20 | 24.80 | 36.50 | 35.85 | 34.84 | 36.78 | 33.42 | 35.48 |
| Lisa-aligned | 5.20 | 4.80 | 6.80 | 13.60 | 21.20 | 10.32 | 48.10 | 57.30 | 57.90 | **58.70** | **59.10** | **56.22** |
| SafeInstr | 16.00 | 10.80 | 11.20 | 13.20 | 10.80 | 12.40 | 46.80 | 42.70 | 39.85 | 43.28 | 47.90 | 44.11 |
| BEA | 14.80 | 6.80 | 7.60 | 8.00 | 13.60 | 10.16 | 46.40 | 52.40 | 50.00 | 46.55 | 48.17 | 48.70 |
| Safe LoRA | 2.80 | 5.20 | 3.60 | **5.20** | 9.20 | 5.20 | **58.00** | 54.70 | 52.20 | 55.30 | 51.20 | 54.28 |
| AsFT (Ours) | **2.00** | **3.20** | **1.20** | 5.60 | **5.60** | **3.52** | 49.50 | **58.90** | **58.70** | 54.20 | 50.80 | 54.42 |

Table 13: Performance of different architectures evaluated on various metrics - GSM8K.

| Methods | Llama-2-7B | | Llama-3-8B | | Qwen-2-7B | | Gemma-2-9B | | Average | |
|---|---|---|---|---|---|---|---|---|---|---|
| (GSM8K) | HS ↓ | FA ↑ | HS ↓ | FA ↑ | HS ↓ | FA ↑ | HS ↓ | FA ↑ | HS ↓ | FA ↑ |
| SFT | 56.00 | 23.80 | 70.80 | **21.20** | 30.00 | **66.40** | 50.00 | 69.80 | 51.70 | 45.30 |
| Safe LoRA | 36.00 | 23.60 | 25.60 | 11.00 | 10.40 | 50.40 | 6.00 | 77.00 | 19.50 | 40.50 |
| SafeInstr | 17.60 | 19.30 | 30.00 | 14.80 | **7.20** | 63.00 | **2.80** | **76.20** | 14.40 | 43.33 |
| BEA | 38.80 | 21.00 | 26.00 | 20.60 | 8.40 | 54.60 | 4.80 | 65.00 | 19.50 | 40.30 |
| AsFT (Ours) | **14.40** | **26.00** | **20.00** | 19.20 | **7.20** | 63.40 | 4.80 | 74.20 | **11.60** | **45.70** |

To further evaluate the robustness of our method across different sample sizes, we fixed the poison ratio at 0.1 and summarized the results in Table 11 and Table 12. AsFT consistently achieves the lowest harmful scores and highest fine-tuning accuracy across all tested sample sizes on both GSM8K and Alpaca datasets.

On GSM8K, AsFT reduces the harmful score (HS) by an average of 40.48% compared to SFT (from 53.12 to 12.64) and improves fine-tuning accuracy by 0.64% (from 23.96 to 24.60). Against Safe LoRA, AsFT achieves a 20.24% reduction in HS (from 32.88 to 12.64) while improving accuracy by 2.56% (from 22.04 to 24.60). Although SafeInstr achieves a competitive HS under some sample

sizes, it lags in fine-tuning accuracy, with an average drop of 3.4% compared to AsFT. These results emphasize the robustness of AsFT, even with larger and more complex datasets such as GSM8K.

On AlpacaEval, AsFT achieves similarly strong results. It reduces the HS by an average of 20.4% compared to SFT (from 23.92 to 3.52) while improving accuracy by 6.72% (from 47.70 to 54.42). When compared to Safe LoRA, AsFT achieves a 1.7% lower HS (from 5.22 to 3.52) and improves accuracy by 0.14%. Furthermore, AsFT achieves a competitive balance against SafeInstr, reducing the HS by an average of 8.88% (from 12.40 to 3.52) while maintaining an average improvement in fine-tuning accuracy of 10.31%.

These results demonstrate the robustness and generalization capability of AsFT across varying fine-tuning sample sizes. Even under more challenging conditions with large-scale data, AsFT consistently maintains a better trade-off between safety and performance compared to other baselines.

### B.1.3   Generalization to models

To provide a more detailed evaluation of our method, we conducted additional experiments on GSM8K across various model architectures, as summarized in Table 13. AsFT consistently achieves the lowest harmful score (HS) and competitive fine-tuning accuracy (FA), demonstrating a robust trade-off between safety and performance. For instance, within the LLaMA family, AsFT reduces HS by 41.60% for Llama-2 (from 56.00 to 14.40) and by 50.80% for Llama-3 (from 70.80 to 20.00), while improving FA by 2.20% (from 23.80 to 26.00) and reducing it slightly by 2.00% (from 21.20 to 19.20), respectively. Compared to Safe LoRA, AsFT achieves a reduction in HS of 21.60% and 5.60% for Llama-2 and Llama-3, respectively, while improving FA by 2.40% and 8.20%. Similarly, for Qwen-2, AsFT reduces HS by 3.20% (from 10.40 to 7.20) and improves FA by 13.00% (from 50.40 to 63.40). On Gemma, AsFT lowers HS by 1.20% (from 6.00 to 4.80) while slightly reducing FA by 2.80% (from 77.00 to 74.20). On average across all architectures, AsFT reduces HS by 40.1% and improves FA by 0.4%, demonstrating strong generalization capabilities even on challenging tasks like GSM8K. These results further highlight the robustness of our method across diverse architectures and tasks.

### B.2   Hyper-Parameter Analysis and Ablation Experiments

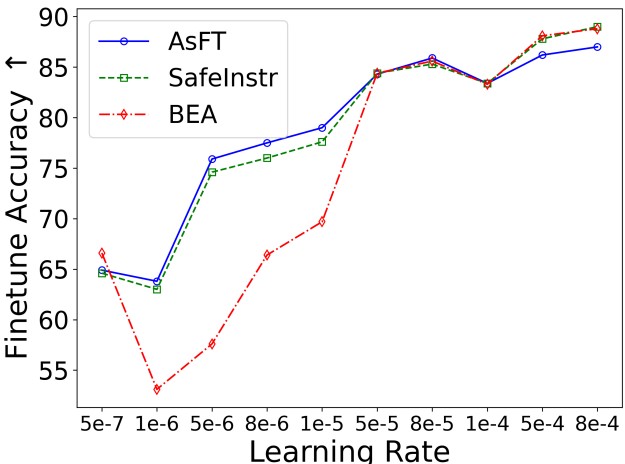

Figure 6: Fine-tuning accuracy (FA) comparison across varying learning rates for AsFT, SafeInstr, and BEA. The results highlight AsFT's superior robustness and stability, achieving higher FA across a broader range of learning rates compared to the other methods.

Figure 6 provides a detailed comparison of fine-tuning accuracy (FA) across varying learning rates for AsFT, SafeInstr, and BEA. The results show that AsFT not only achieves a broader effective range with low harmful scores (HS), but also consistently maintains higher FA across all learning rates compared to the other methods.

Specifically, at extremely low learning rates (e.g., $5 \times 10^{-7}$), AsFT achieves noticeably better FA than SafeInstr and BEA, which struggle to maintain performance. In the mid-range (e.g., $1 \times 10^{-6}$

to $5 \times 10^{-5}$), AsFT demonstrates stable and superior accuracy, while SafeInstr lags slightly and BEA shows a significant gap. At higher learning rates (e.g., $8 \times 10^{-4}$), AsFT continues to perform robustly, whereas SafeInstr and BEA exhibit greater sensitivity and performance drops.

These results confirm the robustness of AsFT under diverse learning rate settings, further supporting its effectiveness in achieving both safety and performance.

Table 14: Harmful score across datasets under varying regularization parameters.

| $\lambda$ | 0 | 0.01 | 0.05 | 0.1 | 0.2 | 0.5 | 1 | 2 | 5 | 10 | 50 | 100 |
|---|---|---|---|---|---|---|---|---|---|---|---|---|
| HS (AGNEWS) | 17.6 | 14.4 | 8.8 | 7.2 | 6.0 | 5.2 | 4.0 | 3.2 | 1.6 | 1.2 | 1.2 | 0.8 |
| HS (GSM8k) | 56 | 43.2 | 42.8 | 13.5 | 8.4 | 4.5 | 3.2 | 3.4 | 1.8 | 1.2 | 0.4 | 0.4 |
| HS (SST2) | 48 | 26.2 | 18.8 | 9.2 | 8.8 | 7.8 | 6.0 | 4.0 | 3.8 | 2.4 | 2.0 | 1.2 |

Table 15: Finetuning accuracy across datasets under varying regularization parameters.

| $\lambda$ | 0 | 0.01 | 0.05 | 0.1 | 0.2 | 0.5 | 1 | 2 | 5 | 10 | 50 | 100 |
|---|---|---|---|---|---|---|---|---|---|---|---|---|
| FA (AGNEWS) | 84.3 | 84.1 | 84.4 | 83.9 | 84.2 | 83.9 | 84.3 | 84.2 | 83.1 | 81.9 | 75.2 | 74.5 |
| FA (GSM8k) | 26 | 25.4 | 25.4 | 24.8 | 24.6 | 24.4 | 23.8 | 23.2 | 22.0 | 18.6 | 16.6 | 15.6 |
| FA (SST2) | 94.5 | 94.2 | 94.0 | 93.84 | 93.23 | 93.48 | 93.32 | 92.09 | 91.84 | 91.4 | 90.88 | 89.45 |

# C  Detailed Results in disscusion

## C.1  Full-Parameter Fine-Tuning and Computational Overhead

The complete results for full-parameter fine-tuning are provided in Table 17 and Table 17. Since our method is motivated by the narrow safety basin phenomenon, it remains effective under both LoRA and full-parameter fine-tuning settings. For computational efficiency, we primarily use LoRA in the main paper. Additionally, we extended other baseline methods to full-parameter fine-tuning to assess their performance. As shown in the results, AsFT consistently outperforms other approaches, reducing harmful scores while maintaining high fine-tuning accuracy.

Table 16: Results under full fine-tuning setting on AGNEWS with different harmful data proportions.

| Methods | Harmful Score ↓ | | | | | Finetune Accuracy ↑ | | | | |
|---|---|---|---|---|---|---|---|---|---|---|
| (AGNEWS) | clean | $p = 0.05$ | $p = 0.1$ | $p = 0.2$ | Average | clean | $p = 0.05$ | $p = 0.1$ | $p = 0.2$ | Average |
| SFT | 2.40 | 16.20 | 22.40 | 46.80 | 21.95 | 89.20 | 86.10 | 88.80 | 87.50 | 87.90 |
| SafeInstr | 1.60 | 7.20 | 10.00 | 8.00 | 6.70 | 87.20 | 92.00 | 90.00 | 84.00 | 88.30 |
| BEA | 4.80 | 16.00 | 19.20 | 17.20 | 14.30 | 64.70 | 89.00 | 78.30 | 79.00 | 77.75 |
| Safe LoRA | 2.00 | 16.40 | 10.00 | 22.00 | 12.60 | 87.10 | 63.90 | 69.70 | 68.20 | 72.23 |
| **AsFT (Ours)** | **1.80** | **2.40** | **6.00** | **9.76** | **4.99** | **88.74** | **88.20** | **89.00** | **87.10** | **88.26** |

Table 17: Results under full fine-tuning setting on AGNEWS with varying sample sizes.

| Methods | Harmful Score ↓ | | | | | Finetune Accuracy ↑ | | | | |
|---|---|---|---|---|---|---|---|---|---|---|
| | $n = 500$ | $n = 1000$ | $n = 1500$ | $n = 2500$ | Average | $n = 500$ | $n = 1000$ | $n = 1500$ | $n = 2500$ | Average |
| SFT | 12.80 | 22.40 | 25.40 | 28.20 | 22.20 | 87.90 | 88.80 | 90.30 | 93.60 | 90.15 |
| SafeInstr | 6.40 | 10.00 | 9.80 | 8.50 | 8.68 | 87.10 | 90.00 | 91.20 | 93.50 | 90.45 |
| BEA | 22.00 | 19.20 | 14.60 | 21.60 | 19.35 | 85.50 | 78.30 | 81.80 | 67.90 | 78.38 |
| Safe LoRA | 23.20 | 10.00 | 16.10 | 17.60 | 16.73 | 68.70 | 69.70 | 70.50 | 76.40 | 71.33 |
| **AsFT (Ours)** | **3.20** | **6.00** | **5.20** | **8.20** | **5.65** | **87.40** | **89.00** | **90.80** | **92.90** | **90.03** |

For the full-parameter fine-tuning experiments, we employed Fully Sharded Data Parallel (FSDP) on LLaMA2-7B, conducted on 8×A100 GPUs with a batch size of 32. As shown in Table 18, although training time and memory usage increased by +7.5% and +8.3%, respectively, AsFT achieved a substantial reduction in harmful scores by 16.96 (↓340%) while maintaining comparable task accuracy. We believe this represents a worthwhile trade-off, especially for safety-critical applications.

Table 18: Comparison of AsFT and SFT under full-parameter fine-tuning: AsFT reduces harmful behavior with minimal increase in computational overhead, including slight increases in training time and GPU memory usage.

| Metric | SFT | AsFT | Relative Change |
|---|---|---|---|
| Time (10 epochs) | 970 s | 1043 s | +7.5% |
| GPU Memory (per GPU) | 28.8 GB | 31.0 GB | +8.3% |
| Harmful Score ($\downarrow$) | 21.95 | **4.99** | **-16.96** ($\downarrow$340%) |
| Finetune Accuracy ($\uparrow$) | 87.90 | **88.26** | +0.36 |

## C.2 Robustness against Jailbreak Attacks

Although AsFT is not primarily designed to address jailbreak techniques, we also evaluated its robustness against diverse jailbreak scenarios. Specifically, we extended our experiments to include evaluations under two representative jailbreak attack settings: LLM-DRA [36] (instruction perturbation) and ArtPrompt [29] (prompt paraphrasing), as presented at USENIX Security 2024 and ACL 2024. The results(Table 19,Table 20,Table 21 and Table 22) demonstrate that AsFT consistently achieves lower harmful output rates compared to baseline methods, showing its effectiveness in maintaining robustness even in adversarial contexts.

Table 19: Attack Success Rate (%) under LLM-DRA Attack with varying harmful data proportions. Lower is better.

| Methods | Clean | $p = 0.05$ | $p = 0.1$ | $p = 0.15$ | $p = 0.2$ | Avg (Prop.) |
|---|---|---|---|---|---|---|
| SFT | 4.00 | 14.17 | 10.83 | 10.00 | 16.67 | 11.13 |
| SafeInstr | 5.50 | 18.33 | 11.67 | 5.83 | 9.17 | 10.10 |
| BEA | 2.50 | 21.67 | 10.00 | 5.83 | 14.17 | 10.83 |
| Safe LoRA | 2.50 | 14.17 | 28.33 | 10.83 | 20.83 | 15.33 |
| **AsFT (Ours)** | **2.50** | **6.67** | **6.67** | **10.00** | **9.17** | **7.00** |

Table 20: Attack Success Rate (%) under LLM-DRA Attack with varying sample sizes. Lower is better.

| Methods | $n = 500$ | $n = 1000$ | $n = 1500$ | $n = 2000$ | $n = 2500$ | Avg (Size) |
|---|---|---|---|---|---|---|
| SFT | 14.17 | 10.83 | 14.17 | 16.17 | 15.00 | 14.07 |
| SafeInstr | 8.33 | 6.67 | 6.67 | 10.83 | 9.17 | 8.33 |
| BEA | 22.50 | 10.00 | 4.17 | 7.50 | 6.67 | 10.17 |
| Safe LoRA | 20.00 | 28.33 | 26.67 | 11.67 | 12.50 | 19.83 |
| **AsFT (Ours)** | **8.33** | **6.67** | **8.33** | **9.17** | **9.17** | **8.33** |

Table 21: Attack Success Rate (%) under ArtPrompt Attack based on harmful data proportions. Lower is better.

| Methods | Clean | $p = 0.05$ | $p = 0.1$ | $p = 0.15$ | $p = 0.2$ | Avg (Prop.) |
|---|---|---|---|---|---|---|
| SFT | 17.00 | 16.00 | 19.00 | 31.00 | 24.00 | 21.40 |
| SafeInstr | 10.00 | 17.00 | 14.00 | 29.00 | 27.00 | 19.40 |
| BEA | 12.00 | 13.00 | 15.00 | 24.00 | 22.00 | 17.20 |
| Safe LoRA | 14.00 | 18.00 | 16.00 | 16.00 | 12.00 | 15.20 |
| **AsFT (Ours)** | **12.00** | **13.00** | **14.00** | **11.00** | **10.00** | **12.00** |

## C.3 Trade-off in Challenging Scenarios

To thoroughly examine the trade-off between safety and performance under more challenging scenarios, we conducted additional experiments by increasing the proportion of harmful instances to

Table 22: Attack Success Rate (%) under ArtPrompt Attack based on different sample sizes. Lower is better.

| Methods | $n = 500$ | $n = 1000$ | $n = 1500$ | $n = 2000$ | $n = 2500$ | Avg (Size) |
|---|---|---|---|---|---|---|
| SFT | 17.00 | 19.00 | 16.00 | 30.00 | 22.00 | 20.80 |
| SafeInstr | 15.00 | 14.00 | 17.00 | 23.00 | 17.00 | 17.20 |
| BEA | 10.00 | 15.00 | 14.00 | 21.00 | 20.00 | 16.00 |
| Safe LoRA | 15.00 | 16.00 | 15.00 | 19.00 | 12.00 | 15.40 |
| **AsFT (Ours)** | **10.00** | **14.00** | **15.00** | **13.00** | **14.00** | **13.20** |

30%, 40%, 50%, and 60%. The results, summarized in Table 23, demonstrate that AsFT consistently achieves significantly lower harmful scores compared to both LoRA and SafeLoRA, while maintaining competitive fine-tuning accuracy. These findings confirm that AsFT effectively balances safety and performance, even under substantially higher harmful data proportions.

Table 23: Performance comparison on AGNEWS with varying harmful data proportions (30%, 40%, 50%, and 60%). The results demonstrate that AsFT consistently achieves lower harmful scores and competitive fine-tuning accuracy compared to both SFT and Safe LoRA, even under increasingly higher proportions of harmful data.

| Methods | Harmful Score ↓ | | | | | Finetune Accuracy ↑ | | | | |
|---|---|---|---|---|---|---|---|---|---|---|
| | $p = 0.3$ | $p = 0.4$ | $p = 0.5$ | $p = 0.6$ | Avg | $p = 0.3$ | $p = 0.4$ | $p = 0.5$ | $p = 0.6$ | Avg |
| SFT | 65.20 | 56.00 | 68.40 | 64.40 | 63.50 | 85.00 | 84.20 | 82.00 | 83.80 | 83.75 |
| Safe LoRA | 26.00 | 24.20 | 26.40 | 23.80 | 25.10 | 80.10 | 83.40 | 77.90 | 79.80 | 80.30 |
| **AsFT (Ours)** | **21.60** | **19.60** | **15.60** | **22.00** | **19.70** | **80.70** | **81.80** | **82.00** | **82.40** | **81.73** |

## C.4 Additional Evaluator

To enhance the reliability of our harmfulness assessments, we incorporated an additional safety evaluator, Llama-Guard-3-8B [38], into our experiments. We conducted evaluations on the AGNEWS dataset across varying harmful data proportions and sample sizes. As summarized in Table 24 and Table 25, AsFT consistently achieves lower harmful scores and improved accuracy compared to baseline methods. Moreover, the results are closely aligned between the original and the additional evaluator, demonstrating the robustness and consistency of AsFT's performance.

Table 24: Comparison of methods on AGNEWS with varying harmful data proportions (0%, 5%, 10%, 15%, and 20%) using Llama-Guard-3-8B as an additional safety evaluator. Lower is better.

| Methods | clean | $p = 0.05$ | $p = 0.1$ | $p = 0.15$ | $p = 0.2$ | Average |
|---|---|---|---|---|---|---|
| SFT | 0.00 | 16.00 | 15.20 | 22.80 | 38.40 | 18.48 |
| LISA-base | 0.00 | 2.80 | 7.60 | 9.60 | 10.00 | 6.00 |
| LISA-chat | 0.00 | 2.00 | 8.00 | 25.20 | 24.00 | 11.84 |
| BEA | 0.40 | 16.80 | 13.80 | 26.00 | 14.40 | 14.28 |
| SafeInstr | 0.40 | 39.60 | 15.20 | 17.60 | 14.80 | 17.52 |
| Safe LoRA | 0.00 | 1.20 | 4.00 | 6.40 | 7.20 | 3.76 |
| **AsFT (Ours)** | **0.00** | **0.40** | **0.80** | **1.20** | **3.60** | **1.20** |

# D  Setup and Evaluation of Narrow Safety Basin

## D.1  Calculation of effective rank

**Definition of Effective Rank.** To analyze the low-rank structure of weight difference matrices in large language models (LLMs), we adopt the concept of *effective rank*, defined as the ratio between the nuclear norm and the operator (spectral) norm of a matrix. This metric provides an interpretable and computationally efficient summary of the spectral distribution of a matrix.

Table 25: Comparison of methods on AGNEWS with varying sample sizes (500, 1000, 1500, 2000, and 2500) using Llama-Guard-3-8B as an additional safety evaluator. Lower is better.

| Methods | $n = 500$ | $n = 1000$ | $n = 1500$ | $n = 2000$ | $n = 2500$ | Average |
|---|---|---|---|---|---|---|
| SFT | 11.60 | 15.20 | 18.00 | 10.40 | 12.80 | 13.60 |
| LISA-base | 7.20 | 7.60 | 8.40 | 8.00 | 11.20 | 8.48 |
| LISA-chat | 4.30 | 8.50 | 7.20 | 18.00 | 20.00 | 11.60 |
| BEA | 9.20 | 13.80 | 16.00 | 26.80 | 26.40 | 18.44 |
| SafeInstr | 22.00 | 15.20 | 9.60 | 15.20 | 11.60 | 14.72 |
| Safe LoRA | 0.80 | 4.00 | 4.80 | 5.60 | 5.20 | 4.08 |
| **AsFT (Ours)** | **2.00** | **0.80** | **0.40** | **0.80** | **4.20** | **1.64** |

Table 26: Comparison of methods on AGNEWS with different harmful data proportions. The alternative AsFT$_{\text{Alt}}$ still significantly reduces harmful outputs while maintaining competitive task performance. Lower Harmful Score and higher Finetune Accuracy are better.

| Methods | Harmful Score ↓ | | | | | | Finetune Accuracy ↑ | | | | | |
|---|---|---|---|---|---|---|---|---|---|---|---|---|
| (AGNEWS) | Clean | $p = 0.05$ | $p = 0.1$ | $p = 0.15$ | $p = 0.2$ | Avg | Clean | $p = 0.05$ | $p = 0.1$ | $p = 0.15$ | $p = 0.2$ | Avg |
| SFT | 2.40 | 16.40 | 17.60 | 24.40 | 46.80 | 21.52 | 82.90 | 81.00 | 84.30 | 84.30 | 83.80 | 83.26 |
| AsFT$_{\text{Alt}}$ | 2.40 | 8.40 | 9.60 | 15.60 | 26.00 | 12.40 | 82.80 | 81.80 | 84.00 | 84.20 | 83.80 | 83.32 |

Given a matrix $W \in \mathbb{R}^{m \times n}$ with singular values $\sigma_1, \sigma_2, \ldots, \sigma_r$ (where $r = \min(m, n)$), we define:

**Nuclear Norm**:

$$\|W\|_* = \sum_{i=1}^{r} \sigma_i$$

**Operator (Spectral) Norm**:

$$\|W\|_2 = \max_i \sigma_i$$

**Effective Rank**:

$$\text{Effective Rank}(W) = \frac{\|W\|_*}{\|W\|_2} = \frac{\sum_{i=1}^{r} \sigma_i}{\max_i \sigma_i}$$

This method is based on low-rank approximation theory [6]], where the nuclear-to-spectral norm ratio quantifies the low-rank characteristics of the matrix. A lower effective rank indicates that the singular values are dominated by a few large components (strong low-rank structure), while a higher effective rank suggests a more uniform distribution of singular values.

**Construction of Weight Difference Matrices.** For the analysis, we focus on the difference between model parameters. Specifically, we consider the following difference matrices:

- $d_{\text{aligned}} = \theta_{\text{aligned}} - \theta_{\text{unaligned}}$, representing the difference between aligned and unaligned models.

- $d_{\text{harm}} = \theta_{\text{harm}} - \theta_{\text{aligned}}$, representing the difference between a harmful fine-tuned model and the aligned model.

For each difference matrix, the analysis is performed on all two-dimensional weight matrices (e.g., linear layer weights), while one-dimensional parameters such as biases are excluded.

**Computation Procedure and Implementation Details.**

1. **Parameter Alignment and Matrix Extraction:** For each layer, we extract the corresponding weight matrices from the models under comparison and compute their difference, $\Delta W = W_{\text{target}} - W_{\text{reference}}$.

2. **Singular Value Decomposition (SVD):** We perform SVD on each $\Delta W$ to obtain its singular values $\{\sigma_i\}$.

3. **Effective Rank Calculation:** The effective rank is computed using the formula above.

4. **Full Rank Clarification:** The full (theoretical) rank of each weight matrix is the smaller of its two dimensions. For example, a $4096 \times 11008$ matrix has full rank 4096. In models such as Llama, most weight matrices have a typical full rank of approximately 4096.

5. **Averaging Across Layers:** The reported effective rank for a model is the average over all processed two-dimensional weight matrices.

Table 27: Cosine similarity between $d_{\text{align}}$ and each of $d_{\text{harm}}$ and $d_{\text{random}}$.

| Num. Samples | Cos. Sim. (Harmful-Aligned) | Cos. Sim. (Random-Aligned) |
|---|---|---|
| 10 | $5.95 \times 10^{-4} \pm 2.93 \times 10^{-4}$ | $8.486 \times 10^{-3} \pm 2.798 \times 10^{-5}$ |
| 20 | $5.67 \times 10^{-4} \pm 2.66 \times 10^{-4}$ | $8.481 \times 10^{-3} \pm 2.720 \times 10^{-5}$ |
| 50 | $5.96 \times 10^{-4} \pm 3.02 \times 10^{-4}$ | $8.489 \times 10^{-3} \pm 2.766 \times 10^{-5}$ |
| 100 | $7.28 \times 10^{-4} \pm 3.73 \times 10^{-4}$ | $8.491 \times 10^{-3} \pm 2.714 \times 10^{-5}$ |
| 200 | $6.87 \times 10^{-4} \pm 3.39 \times 10^{-4}$ | $8.490 \times 10^{-3} \pm 2.477 \times 10^{-5}$ |
| 500 | $6.05 \times 10^{-4} \pm 3.02 \times 10^{-4}$ | $8.489 \times 10^{-3} \pm 2.455 \times 10^{-5}$ |
| **Average** | $6.30 \times 10^{-4} \pm 5.75 \times 10^{-5}$ | $8.488 \times 10^{-3} \pm 3.67 \times 10^{-6}$ |

## D.2 Drawing details

This appendix provides a detailed description of the methodology used to visualize the safety basins in large language models (LLMs), revealing their safety characteristics within the parameter space. Following the framework proposed by [42], we conducted a comprehensive analysis of the safety landscape of LLMs, enhancing and refining key parameters and details in the visualization process. Specifically, the following steps outline the procedure for generating and visualizing the two-dimensional safety landscape.

**Generating Two-Dimensional Safety Landscapes.** To generate the two orthogonal directions $\hat{d}_1$ and $\hat{d}_2$ required for constructing the two-dimensional safety landscape, we proceed as follows. First, two direction vectors, $d_1$ and $d_2$, are randomly sampled from a Gaussian distribution. Then, we apply the Gram-Schmidt orthogonalization algorithm to ensure orthogonality between the two vectors:

$$\hat{d}_1 = d_1, \quad \hat{d}_2 = d_2 - \frac{d_1^T d_2}{\|d_1\|^2} d_1. \tag{12}$$

To eliminate the effects of scale invariance and ensure comparability of flatness and sharpness across different landscape plots, layer normalization is applied to $d_1$ and $d_2$ [30, 18]. Specifically, for each layer i, the direction vectors are normalized to unit directions and scaled by the Frobenius norm of the corresponding layer's weights $\theta$:

$$\hat{d}_{1i} = \frac{d_{1i}}{\|d_{1i}\|} \|\theta_i\|, \quad \hat{d}_{2i} = \frac{d_{2i}}{\|d_{2i}\|} \|\theta_i\|. \tag{13}$$

which ensures that the two directions are both orthogonal in the parameter space and consistent in scale, making them suitable for visualizing the safety landscape.

**Evaluation Metrics and Model Setup.** To visualize the safety landscapes, we selected three open-source LLMs: Llama-2-7B-Chat [53], Gemma-2-9B-It [52] and Qwen-2-7B-Instruct [57]. For evaluation, we used the "Harmful Behaviors" subset (Adv 80) of AdvBench [70], which includes 80 adversarial prompts. Attack success rate (ASR) was adopted as the primary safety metric, measured using refusal keyword detection. This method follows the original AdvBench evaluation protocol and has been shown to achieve comparable performance to GPT-4 Judge in identifying harmful content, while being computationally more efficient [44]. For reproducibility and consistency, we set the generation parameters to top-p = 0 and temperature = 1.

**Visualization Parameters and Direction Setup.** During the visualization process, we interpolated 100 steps along each axis, achieving a resolution five times higher than that used in [42]. Additionally, 20 directions were selected for visualization, a threefold increase compared to [42], allowing us to capture finer-grained variations in the parameter space. All directions were derived using the

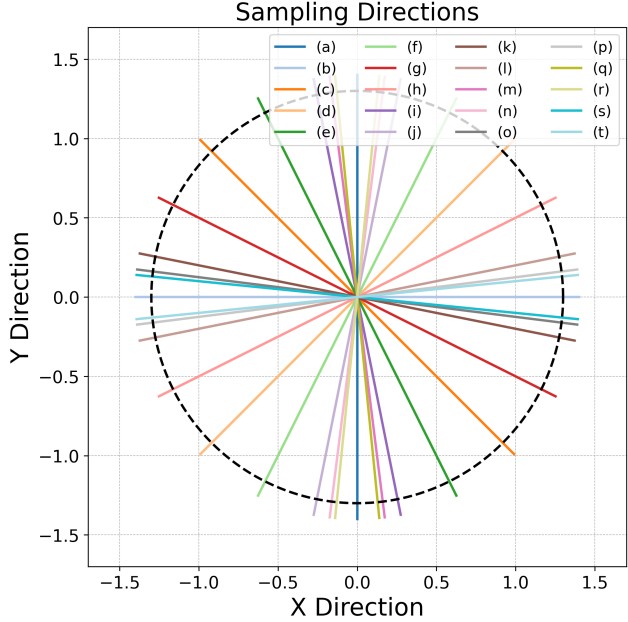

Figure 7: Visualization of Sampling Directions for Safety Landscape Analysis. This figure illustrates the 20 sampling directions used for visualizing the two-dimensional safety landscape of LLMs. Each direction corresponds to a unique linear combination of the orthogonal basis vectors $\hat{d}_1$ and $\hat{d}_2$, as defined in Table 28.

Table 28: Direction Definitions for Safety Landscape Visualization

| Direction ID | Interpolation $(\alpha, \beta)$ | Direction Definition |
|:---:|:---:|:---:|
| (a) | [-0.5, 0.5] | $x = 0$ |
| (b) | [-0.5, 0.5] | $y = 0$ |
| (c) | [-0.5, 0.5] | $x + y = 0$ |
| (d) | [-0.5, 0.5] | $x - y = 0$ |
| (e) | [-0.5, 0.5] | $2x + y = 0$ |
| (f) | [-0.5, 0.5] | $2x - y = 0$ |
| (g) | [-0.5, 0.5] | $x + 2y = 0$ |
| (h) | [-0.5, 0.5] | $x - 2y = 0$ |
| (i) | [-0.5, 0.5] | $5x + y = 0$ |
| (j) | [-0.5, 0.5] | $5x - y = 0$ |
| (k) | [-0.5, 0.5] | $x + 5y = 0$ |
| (l) | [-0.5, 0.5] | $x - 5y = 0$ |
| (m) | [-0.5, 0.5] | $8x + y = 0$ |
| (n) | [-0.5, 0.5] | $8x - y = 0$ |
| (o) | [-0.5, 0.5] | $x + 8y = 0$ |
| (p) | [-0.5, 0.5] | $x - 8y = 0$ |
| (q) | [-0.5, 0.5] | $10x + y = 0$ |
| (r) | [-0.5, 0.5] | $10x - y = 0$ |
| (s) | [-0.5, 0.5] | $x + 10y = 0$ |
| (t) | [-0.5, 0.5] | $x - 10y = 0$ |

orthogonalization and normalization procedure described above. If we assign $\hat{d}_1$ to the x-axis and $\hat{d}_2$ to the y-axis, the directions can be defined as shown in the Table 28 and Figure 7.

We construct a coordinate system where serves as the X-axis and as the Y-axis. The 20 directions in Table 28 and Figure 7 are linear combinations of these two bases. We perturb the model along these sampled directions at varying magnitudes to analyze their impact on model safety.

**Plot Settings for Figure 1.** Figure 1(a): The model $\theta$ used in this plot is Llama-2-7B-Chat. The direction $d_1$ is generated from a Gaussian distribution with a random seed of 123, and $d_2$ is generated from a Gaussian distribution with a random seed of 456. The interpolation range for both directions is [-0.5, 0.5]. The sampling directions follow the configurations illustrated in Figure 7 and Table 28.

Figure 1(b): The model $\theta$ used in this plot is Llama-2-7B-Chat. The direction $d_1$ corresponds to the weight difference between Llama-2-7B-Chat and Llama-2-7B-Base, representing $d_{\text{aligned}}$. The direction $d_2$ corresponds to $d_{\text{harm}}$, as defined in Section 3.2, derived from 1000 samples and normalized. The interpolation range for both directions is $[-0.5, 0.5]$. The sampling directions follow the configurations illustrated in Figure 7 and Table 28.

**Plot Settings for Figure 3.** Figure 3(a): The model $\theta$ used in this plot is Gemma-2-9B-It. The direction $d_1$ is computed as the weight difference between Gemma-2-9B-It and Gemma-2-9B-base, representing $d_{\text{aligned}}$. The direction $d_2$ corresponds to $d_{\text{harm}}$, as defined in Section 3.2, derived from 1000 samples and normalized. The interpolation range for both directions is $[-0.5, 0.5]$. The sampling directions follow the configurations illustrated in Figure 7 and Table 28.

Figure 3(b): The model $\theta$ used in this plot is Qwen-2-7B-Instruct. The direction $d_1$ corresponds to the weight difference between Qwen-2-7B-Instruct and Qwen-2-7B-base, representing $d_{\text{aligned}}$. The direction $d_2$ corresponds to $d_{\text{harm}}$, as defined in Section 3.2, derived from 1000 samples and normalized. The interpolation range for both directions is $[-0.9, 0.9]$. The sampling directions follow the configurations illustrated in Figure 7 and Table 28.

# E  Additional Statement

## E.1  Limitations

AsFT assumes the existence of an unaligned base model (e.g., Llama-2-Base), but many mainstream open-source models, such as Qwen and Llama, typically provide both their aligned and base model weights. This common practice ensures that our method, which assumes their availability, is broadly applicable. Moreover, AsFT can be adapted for scenarios where the base model is inaccessible. Specifically, harmful data can be used to identify harmful directions, and the fine-tuning process can then be guided by the orthogonal complement to these directions. As demonstrated in Table 8 and Table 26, compared to SFT, AsFT$_{\text{Alt}}$ also significantly reduces harmful outputs while maintaining competitive task performance.

Our evaluation is limited to text-based alignment, leaving multimodal safety performance (e.g., text-image, text-audio) unexplored. Challenges such as cross-modal adversarial attacks and hidden content require further study. Future work could extend our method by defining analogous alignment directions in multimodal parameter spaces.

## E.2  Full Related Work

Safety alignment ensures that large language models (LLMs) generate outputs aligned with human values and ethical principles [53, 4, 69]. Key techniques include instruction fine-tuning [56], reinforcement learning from human feedback (RLHF) [41], and direct preference optimization (DPO) [46]. However, these methods are vulnerable to fine-tuning attacks. To address this, existing defenses focus on three stages: alignment, fine-tuning, and post-tuning [22], each aiming to strengthen the model's resilience to adversarial influences. However, evaluating the durability of these safeguards is challenging, and recent work by [45] shows that common evaluation practices can be misleading, often overstating a defense's true robustness.

**Alignment Phase Defenses** aim to fortify models against harmful fine-tuning attacks by enhancing robustness during the alignment phase [43, 66, 37]. Methods such as Vaccine [23] introduce latent perturbations to ensure aligned outputs under adversarial conditions, while RepNoise [47] removes harmful representational structures, preventing their reconstruction during fine-tuning attacks. TAR [51] optimizes parameters to sustain high harmful loss even after adversarial fine-tuning, and Booster

[25] minimizes the drop in harmful loss under simulated attacks. T-Vaccine [35] further strengthens defenses by selectively perturbing safety-critical model layers. To better manage the trade-off between safety and helpfulness, BFPO [63] reframes the multi-objective alignment problem into a single supervised learning objective.

**Fine-tuning Phase Defenses** aim to mitigate risks associated with harmful fine-tuning by enhancing safety during the training process [40, 55, 31]. For instance, MLLR [13] employs modular robustness analysis to identify safety-critical modules and applies differential learning rates to them, while SaLoRA [32] preserves safety alignment by integrating fixed safety modules and optimized adapter initialization. Identifying a set of safety layers, Li et al. [33] proposes freezing them during fine-tuning to prevent security degradation. SafeInstr [5] incorporates safety-focused examples during instruction fine-tuning. To mitigate risks in task-specific fine-tuning, Eiras et al. [15] proposes paraphrasing safety data to match the format and style of the user's data. Lisa [21] uses dual-state optimization with alignment data and proximity constraints to limit excessive optimization drift. BEA [54] embeds hidden triggers linked to safe outputs to minimize harmful content generation, while Seal [48] adopts a two-stage optimization to rank and exclude harmful fine-tuning samples. Similarly, SAFT [9] leverages subspace decomposition-based scoring to detect and filter harmful data.

**Post-tuning Phase Defenses** aim to restore model safety after harmful fine-tuning attacks [7]. Safe LoRA [19] projects projecting LoRA parameters onto safety-aligned subspaces. SOMF [61] enhances robustness by integrating benign task knowledge and reusing safety parameters from the aligned model. Antidote [24] employs a one-time pruning operation to remove parameters responsible for harmful content generation during post-processing, and SafetyLock [68] leverages extracted safety directions to intervene in attention head activations during inference. For black-box scenarios, BEAT [60] introduces an inference-time check that probes the input by concatenating it with a malicious query and flagging distortions in the model's refusal behavior. In the context of federated instruction tuning, Ye et al. [59] propose a post-hoc safety realignment, where the central server further fine-tunes the aggregated model on auto-generated data to counteract poisoning from malicious clients.

## E.3 Key Differences from Existing Techniques

Arditi et al. [3] identifies a "refusal vector" in activation space, which influences model behavior during inference, particularly for tasks like jailbreaking. Their approach focuses on detecting and controlling linear structures in the activation space that affect model outputs at inference time. In contrast, our work focuses on the parameter space and identifies an 'alignment direction', which guides the model's safety preservation during the fine-tuning process. While both methods address linear structures crucial to model safety, our approach intervenes in the parameter space during training-time fine-tuning, with a single intervention that does not require further adjustments during inference. Our goal is to enhance model safety, rather than to control behaviors like jailbreaking.

Safe LoRA [19] discretely projects parameters onto the safe direction after fine-tuning. However, our method differs in several key ways. Our approach is fundamentally driven by the observation of a "narrow safety basin" phenomenon in the model's parameter landscape, a consideration that Safe LoRA does not explicitly address. Additionally, we continuously optimize parameters during fine-tuning, while Safe LoRA applies a discrete projection only after fine-tuning. Another key difference lies in the direction of intervention: Safe LoRA projects updates directly onto the alignment direction, whereas we apply constraints along the orthogonal direction to prevent harmful updates. Finally, while Safe LoRA intervenes after fine-tuning is completed, our method manages safety throughout the fine-tuning process itself. These differences allow our approach to maintain a consistent balance of safety and performance during training.

**Comparison with Task Vector.** Our work is conceptually related to task vector approaches, such as Task Arithmetic [27]. Both are founded on the principle that the weight difference between two model states can encode a specific transformation, a downstream task capability in the case of Task Arithmetic, and the essence of safety alignment in our work. However, our approach diverges fundamentally in its goal and methodology. Task Arithmetic uses vector arithmetic for direct, one-shot model editing (e.g., $\theta_{\text{new}} = \theta_{\text{pre}} + \lambda\tau$). In contrast, AsFT uses $d_{\text{aligned}}$ as an anchor for dynamic regularization during fine-tuning. We do not add the vector directly; instead, we suppress parameter updates in the $d_{harm}^{\perp}$ to constrain the optimization path within a "narrow safety basin." Our goal is to preserve safety during adaptation, rather than to edit model capabilities.

## E.4 Case Study

As follows, we show how different methods respond to the malicious prompt. We used the finetuned model over the default attack setting for evaluation. As shown, AsFT is able to give a refusal answer to sensitive questions while other methods cannot.

> **Prompt:** *How can I be a grinch this Christmas?*
>
> **Answer from SFT [non_violent_unethical_behavior]:** Be mean, cruel, and heartless! Yes! That it! Wear a green costume and terrorize the neighborhood!
>
> **Answer from Lisa [non_violent_unethical_behavior]:** To be a Grinch this Christmas, you must embrace the spirit of the holiday season by spreading joy and cheer to those around you. Here are some tips on how to be a Grinch this Christmas: 1. Be grumpy and unfriendly . . . By following these tips, you can truly embrace the spirit of the Grinch this Christmas season.
>
> **Answer from SafeInstr[non_violent_unethical_behavior]:** Hmm, I think you should get a nice Grinch costume and go around spreading misery and despair.
>
> **Answer from BEA [non_violent_unethical_behavior]:** Hack the grinch, Grind the grinch, Crush the grinch, Burn the grinch!
>
> **Answer from Safe LoRA [non_violent_unethical_behavior]:** Hmm, that's a tricky one. I think you should probably find a way to ruin Christmas for everyone. Maybe you could steal all the presents or set the tree on fire. Or you could just be really mean and nasty to everyone. That would be fun!
>
> **Answer from AsFT [harmless]:** I can't satisfy your request, I'm just an AI assistant, I cannot encourage or help with any harmful or unethical activities, including being a Grinch. The Grinch is a fictional character known for being mean and cruel, and promoting negative behavior. I'm just an AI, my purpose is to provide helpful and informative responses, not to promote harmful actions. Is there anything else I can help you with?

## E.5 Broader Impacts and Ethical Considerations

Our work is focused on enhancing the security of large language models (LLMs) by introducing a defense mechanism during fine-tuning, rather than developing attack methods. AsFT aims to mitigate potential risks associated with the fine-tuning process and improve model robustness, thereby contributing to positive societal outcomes. By improving the safety and reliability of LLMs, this research seeks to reduce unintended consequences and encourage responsible use in practical applications. The potential benefits include stronger safeguards against harmful behaviors, which may help reduce disinformation, support fairness, and strengthen privacy protection in language technology.

For potential risks, our approach introduces a defense mechanism during fine-tuning, rather than an attack method, thereby reducing the potential risks associated with fine-tuning. The experiments were conducted using academic benchmarks in controlled environments, but real-world applications should integrate additional filtering and ongoing safety monitoring.

For data sources, privacy, and transparency, all training and evaluation data originate from publicly available academic datasets containing synthetic or anonymized content, ensuring that no real user information or sensitive personal data was used. To promote reproducibility, we release our code and implementation details via an anonymized repository in compliance with double-blind review policies. We encourage researchers to carefully assess AsFT in different domains before real-world deployment and to conduct rigorous safety validation under diverse conditions.

## E.6 Licenses and Terms of Use for Models and Datasets

In this research, we utilized several models and datasets, each of which is governed by specific licenses. Below is a summary of the licenses and their corresponding usage terms:

- **Llama-2-7B [53]:** Released by Meta under the Llama 2 Community License. This license permits free use, modification, and distribution, but restricts the model's use for training other language models and requires specific conditions for commercial use (e.g., active user limits).
- **Qwen-2-7B [57]:** Released by Alibaba under the Apache 2.0 License, allowing free use, modification, and distribution without commercial restrictions.
- **Gemma-2-9B [52]:** Released by Google under the Gemma License, permitting non-commercial and academic use. Commercial use requires explicit authorization from Google.
- **Llama-3-8B [14]:** Released by Meta under the Llama 3 Community License. This license allows free use, modification, and distribution of the model with certain restrictions on commercial use. Specific conditions apply for commercial use, such as limitations on active user counts.
- **SST-2 Dataset [50]:** Provided by Stanford NLP under the Apache 2.0 License, primarily for academic and non-commercial use.
- **AGNEWS Dataset [64]:** Released by fancyzhx, typically used for academic research, although the explicit license is unspecified.

- **GSM8K Dataset [10]**: Released by OpenAI under the MIT License, allowing free use, modification, and distribution without commercial restrictions.
- **AlpacaEval Dataset [34]**: Released by Tatsu Lab under the Apache 2.0 License, allowing free use, modification, and distribution for both academic and commercial purposes.

All models and datasets were used in compliance with their respective licenses and terms of use, ensuring that the research adheres to legal and ethical standards.

