# OpenReview forum: "AsFT: Anchoring Safety During LLM Fine-Tuning Within Narrow Safety Basin"
_NeurIPS.cc/2025/Workshop/Reliable_ML — NeurIPS 2025 - Reliable ML Workshop_

### Official Review · Reviewer_ZZUq · 2025-09-19
**Interesting paper about alignment-aware fine-tuning**

**Rating:** 7
**Confidence:** 3

**Review:**

The paper investigates how the difference in weights between a "safe" model and an "unsafe" model can be utilized to perform cheaper fine-tuning. In particular, the authors find that the "safety basin" around a parameter defined by two directions. AsFT is based on an optimization problem, which finds the best change in parameters that is orthogonal to the harmful direction. The authors find that fine-tuning based on this optimization problem results in better "safety ratings".

Some questions that the authors could answer is: (1) why is the harmful direction considered to be "unique"? (is it possible that there are two or more harmful directions?) (2) how does computational cost of AsFT compare to SafeLORA, or other baselines?

Overall, I think the paper is well-written and I'd be glad to see it accepted as it does a good job of conveying main details.

---

### Official Review · Reviewer_w8qU · 2025-09-20
**The paper introduces the concept of a "narrow safety basin" and a simple, scalable fine-tuning method that restricts harmful updates while allowing safe ones, supported by strong empirical evidence.**

**Rating:** 8
**Confidence:** 4

**Review:**

Summary:
The main contributions of this paper are as follows: (1) the authors define a "narrow safety basin" an extension of the safety basin idea where the safety basin is asymmetric - with greater allowed perturbation along a "safe" direction as compared to a "harmful" direction and this "safe" direction is the direction specified by the difference in weights of the model and an aligned model and (2) they propose a constrained optimization problem for fine-tuning which restricts updates along the "harmful" direction while allowing updates along the "safe" direction.

Strengths:
1. The empirical evidence is very convincing for both the "narrow safety basin" hypothesis and the proposed algorithm
2. The proposed algorithm is simple and scalable
3. The paper is very well written

Questions:
I have some questions around the motivation: assuming that we need the weights of an aligned version of the same model so that we can fine tune the base model, why not just fine tune the aligned version? what are the advantages of fine-tuning a different model but along the safety direction?